# Effector loading onto the VgrG carrier activates type VI secretion system assembly

Chih-Feng Wu[1,2,§] (ID), Yun-Wei Lien[1,3,4,§] (ID), Devanand Bondage[1,†] (ID), Jer-Sheng Lin[1,‡], Martin Pilhofer[4] (ID),
Yu-Ling Shih[5] (ID), Jeff H Chang[2,6] (ID) & Erh-Min Lai[1,3,*] (ID)

## Abstract

The type VI secretion system (T6SS) is used by many bacteria to engage in social behavior and can affect the health of its host plant or animal. Because activities associated with T6SSs are often costly, T6SSs must be tightly regulated. However, our knowledge regarding how T6SS assembly and contraction are regulated remains limited. Using the plant pathogen *Agrobacterium tumefaciens*, we show that effectors are not just passengers but also impact on T6SS assembly. The *A. tumefaciens* strain C58 encodes one T6SS and two Tde DNase toxin effectors used as major weapons for interbacterial competition. Here, we demonstrate that loading of Tde effectors onto their cognate carriers, the VgrG spikes, is required for active T6SS secretion. The assembly of the TssBC contractile sheath occurs only in the presence of Tde effectors. The requirement of effector loading for efficient T6SS secretion was also validated in other *A. tumefaciens* strains. We propose that such a mechanism is used by bacteria as a strategy for efficacious T6SS firing and to ensure that effectors are loaded onto the T6SS prior to completing its assembly.

**Keywords** *Agrobacterium tumefaciens*; effector; interbacterial competition; type VI secretion system; VgrG

**Subject Categories** Microbiology, Virology & Host Pathogen Interaction; Structural Biology

## Introduction

The type VI secretion system (T6SS) is a versatile secretion system that has been implicated in virulence, antagonism, nutrient acquisition, and horizontal gene transfer [1–3]. Contact-dependent interbacterial competition appears to be the major function of T6SS, a function that can influence the composition of microbial communities [4,5]. A T6SS machine resembles a contractile phage tail-like structure [6–8]. Effectors are loaded onto the T6SS either via non-covalent interactions (cargo effectors) or as carboxy-terminal extensions to either of the three core structural components (Hcp, VgrG, or PAAR) [2,9]. Upon contraction, the puncturing device loaded with effectors is fired and propelled, carrying effectors, across the cell membrane into the extracellular milieu or into targeted bacterial or eukaryotic cells. In general, cargo effectors are considered as accessory components of the secretion apparatus. This is based on the repeated observations that Hcp and/or VgrG, key markers for T6SS secretion, can be detected in the extracellular milieu of mutants lacking effector genes [10–14].

The plant pathogenic bacterium *Agrobacterium tumefaciens* strain C58 encodes one T6SS main cluster consisting of the *imp* and *hcp* operons and *vgrG2* operon distal to the main cluster [15]. Three T6SS toxin effectors were identified, in which secretion of Tde1 and Tde2 DNases is governed specifically by VgrG1 and VgrG2, respectively, and secretion of Tae amidase is likely mediated by Hcp [16,17]. Tde effectors are major weapons deployed by *A. tumefaciens* for interbacterial competition *in planta* [10]. Each of the effector genes is genetically linked to a cognate immunity gene, which form three toxin–immunity pairs (i.e., *tae-tai* and *tde1-tdi1* located in the main gene cluster and *tde2-tdi2* encoded downstream of *vgrG2* distal to the main cluster; Fig 1A). Self-intoxication is prevented in the toxin-producing cells by the cognate immunity proteins.

Here, we show that the loading of cargo effectors onto their cognate VgrG spike proteins is required for efficient T6SS-dependent secretion by *A. tumefaciens*. We demonstrate that in the absence of VgrG cargo effector genes, the levels of secretion of Hcp and VgrG and formation of the TssBC sheath are significantly reduced in *A. tumefaciens*. Given the prevalence of T6SS-encoding loci in host-associated bacteria, these findings inform on mechanisms that may influence the composition of microbial communities.

1 Institute of Plant and Microbial Biology, Academia Sinica, Taipei, Taiwan
2 Department of Botany and Plant Pathology, Oregon State University, Corvallis, OR, USA
3 Department of Plant Pathology and Microbiology, National Taiwan University, Taipei, Taiwan
4 Institute of Molecular Biology & Biophysics, Eidgenössische Technische Hochschule Zürich, Zürich, Switzerland
5 Institute of Biological Chemistry, Academia Sinica, Taipei, Taiwan
6 Center for Genome Research and Biocomputing, Oregon State University, Corvallis, OR, USA
*Corresponding author. Tel: +886 2 27871158; Fax: +886 2 27827954; E-mail: emlai@gate.sinica.edu.tw
§These authors contributed equally to this work
†Present address: Division of Molecular and Cellular Biology, Eunice Kennedy Shriver National Institute of Child Health and Human Development, National Institutes of Health, Bethesda, MD, USA
‡Present address: Department of Organismic Interactions, Max Planck Institute for Terrestrial Microbiology, Marburg, Germany

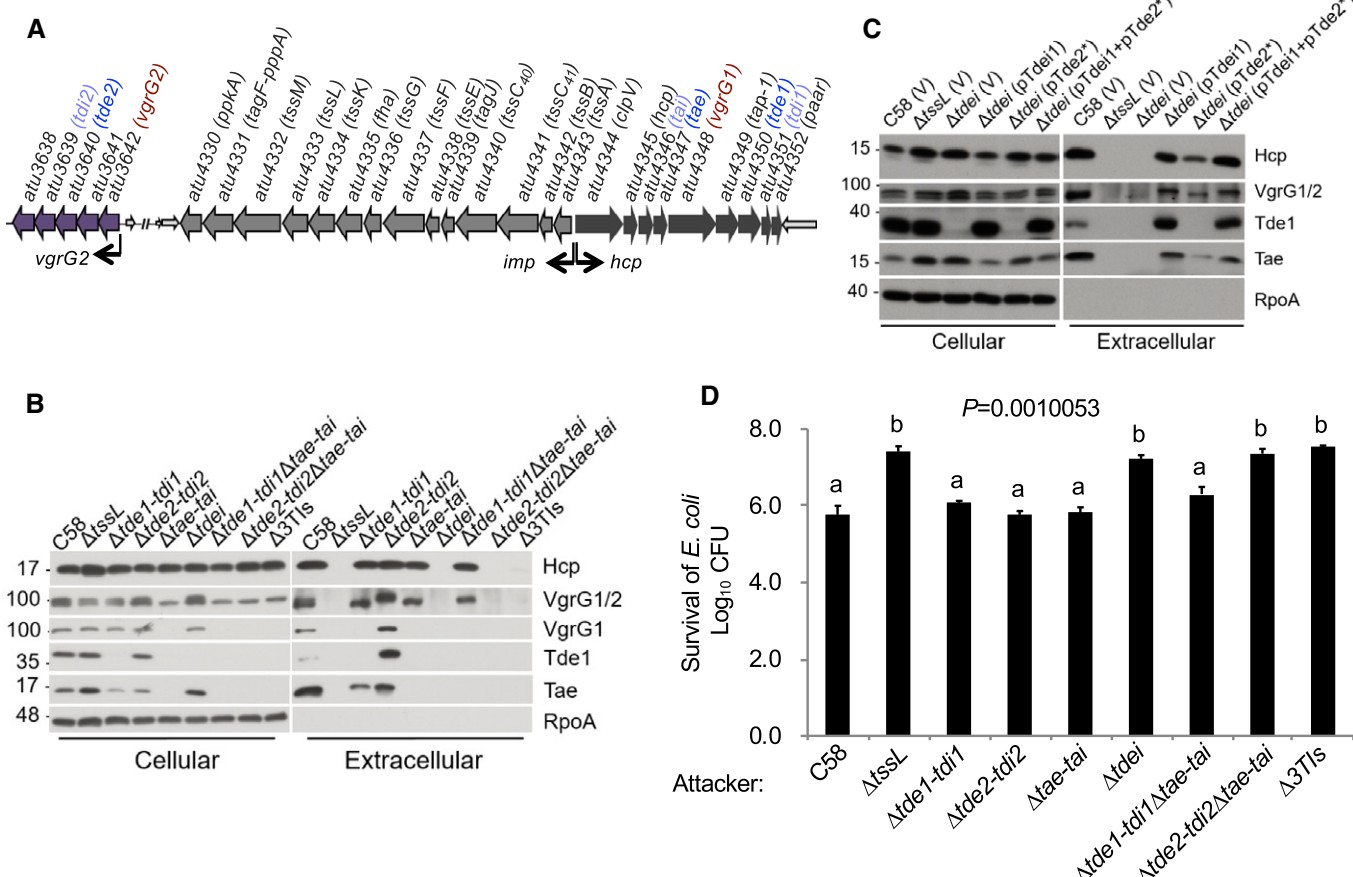

**Figure 1. Presence of Tde effectors in the cell is critical for secretion of the cognate VgrG.**

A   The structure of the *imp* and *hcp* operons and *ugrG2* operon in *Agrobacterium tumefaciens* C58 [10]. The arrows represent coding sequencing, with arrowheads depicting the direction of expression. The wider arrows represent genes in the T6SS-associated operons. The *ugrG2* operon is located distal to *imp* operon and *hcp* operon. The *ugrG* and toxin–immunity gene pairs are highlighted in colors.

B   T6SS secretion assay of *A. tumefaciens* strains: wild-type C58, various mutants lacking one, two, or three toxin–immunity gene pairs, and a mutant lacking *tssL*.

C   T6SS secretion assay of various *A. tumefaciens* strains: wild-type C58, ΔtssL, the *tde* double deletion mutant (Δtdei) containing pRL662 and pTrc200 empty vector (V) only, or expression of pTdei1 (*tde1-tdi1* expressed from pTrc200), pTde2* (catalytic site-mutated *tde2* expressed on pRL662), or pTdei1+ pTde2*.

D   *Agrobacterium tumefaciens* antibacterial activity assay against *Escherichia coli*. The strains of *A. tumefaciens* were co-cultured at a ratio of 30:1 with *E. coli* DH10B (+ pRL662) on LB agar. The survival of target *E. coli* cells was quantified by counting CFUs on gentamicin-containing LB agar plates. Data represent mean ± SEM of 6 biological replicates from three independent experiments. One-way ANOVA followed by Turkey HSD test was used for statistical analysis. Two groups with significant differences (*P* = 0.0010053) are indicated with different letters (a and b).

Data information: In (B, C), cellular and extracellular fractions were collected from *A. tumefaciens* strains grown in liquid 523 medium. Western blots were probed with indicated antibodies; the α-VgrG1 antibody detects VgrG1 (upper band) and VgrG2 (lower band), while α-VgrG1C detects only VgrG1 [16]. RpoA is RNA polymerase subunit alpha, which is localized to the cytosol of *A. tumefaciens*. Molecular weight markers (in kDa) are indicated on the left.
Data are from one independent experiment and reproduced in at least two independent experiments.
Source data are available online for this figure.

# Results

## Tde effector loading onto VgrG is required for secretion of cognate VgrG

Despite evidence suggesting that effectors are not components of the secretion apparatus [10–14], our previous study showed that *A. tumefaciens* strains with variants of VgrG1 lacking the Tde1-binding domain are able to secrete Hcp but at slightly lower levels [16]. This led us to hypothesize that effector-loaded VgrG is more efficiently recruited for T6SS assembly and/or secretion. To test this

hypothesis, we first examined whether the secretion of Hcp and VgrG proteins, a hallmark of T6SS firing, is affected by the presence or absence of effector genes. We found that the secretion of Hcp and VgrG proteins is affected by the presence or absence of Tde effector genes in *A. tumefaciens*. In wild-type C58, but not ΔtssL, a secretion-deficient mutant, Hcp, VgrG1/2, Tde1, and Tae were detected in the medium, referred as extracellular fraction, confirming their T6SS-dependent secretion (Fig 1B). However, VgrG1 and VgrG2 proteins were not detected in the extracellular fraction of mutant strains in which their cognate effector gene was absent. VgrG1 was not detectable in the extracellular fraction of any mutant minimally lacking the

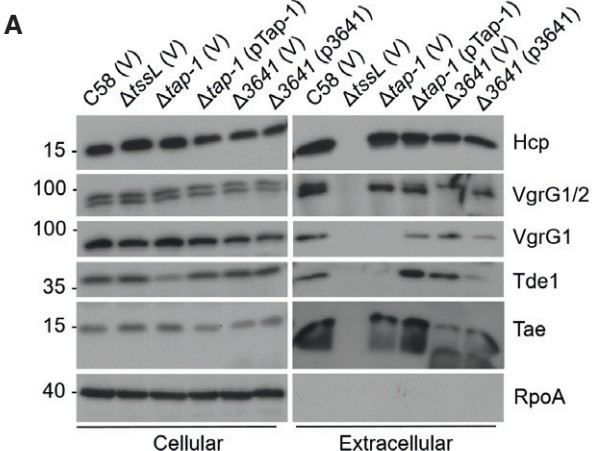

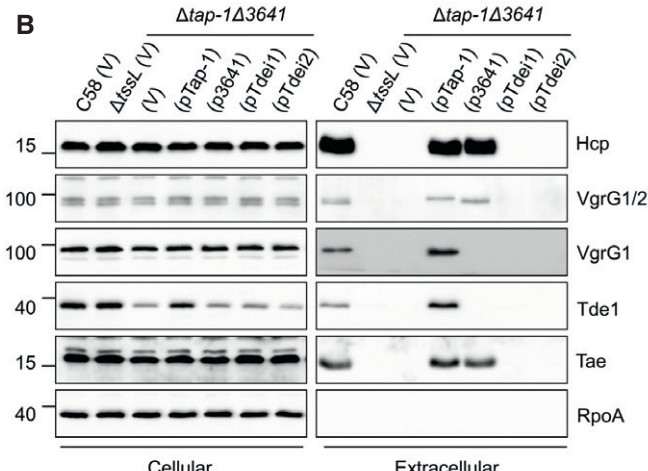

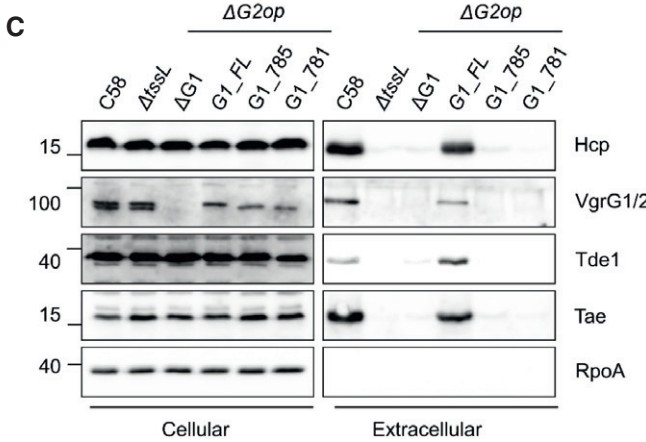

*tde1-tdi1* toxin–immunity gene pair (i.e., Δ*tde1-tdi1*, Δ*tdei* lacking both *tde1-tdi1* and *tde2-tdi2*, and Δ3TIs lacking all three toxin–immunity pairs). Similarly, VgrG2 is inferred to behave in a similar manner in mutants with *tde2-tdi2* toxin–immunity gene pair deleted. Complementing the Δ*tdei* mutant with *tde1-tdi1* restored its ability to secrete VgrG1, while the mutant expressing *tde2\** (encoding Tde2 variant with catalytic site mutation) failed to secrete VgrG1. In contrast, the

**Figure 2. Tde effector loading onto VgrG is required for secretion of the cognate VgrG.**

A  T6SS secretion assay of *Agrobacterium tumefaciens* strains: wild-type C58, Δ*tssL*, Δ*tap-1*, and Δ*atu3641* harboring a pTrc200 vector (V) or derivatives pTap-1 (*tap-1* expressed from pTrc200), or p3641 (*atu3641* expressed from pTrc200).

B  T6SS secretion assay of *A. tumefaciens* C58 Δ*tap-1*Δ*atu3641* containing an empty pTrc200 vector (V) or its derivatives expressing *tap-1, atu3641, tde1-tdi1*, or *tde2-tdi2*.

C  T6SS secretion assay of *A. tumefaciens* C58Δ*G2op* mutant encoding full-length or truncated VgrG1 proteins.

Data information: In (A–C), cellular and extracellular fractions were collected from *A. tumefaciens* strains grown in liquid 523 medium. Western blots were probed with indicated antibodies; the α-VgrG1 antibody detects VgrG1 (upper band) and VgrG2 (lower band), while α-VgrG1C detects only VgrG1 [16]. RpoA is RNA polymerase subunit alpha, which is localized to the cytosol of *A. tumefaciens*. Molecular weight markers (in kDa) are indicated on the left. Data are from one independent experiment and reproduced in at least two independent experiments.

Source data are available online for this figure.

mutant carrying *tde2\**, but not *tde1-tdi1*, restored the secretion of VgrG2 (Fig 1C), which also suggested that DNase activity of Tde effectors is not required for T6SS assembly and secretion. The levels of Hcp were similar in the extracellular fractions of the wild-type strain as well as that from each of the mutants lacking a single toxin–immunity gene pair. Interbacterial competition assays showed that *A. tumefaciens* C58 can only kill *Escherichia coli* when at least one Tde effector is delivered (Fig 1D). The reliance on Tde DNases but not Tae amidase as the primary effectors against *E. coli* is consistent with previous finding that Tde but not Tae influences the *in planta* interbacterial competition activity of *A. tumefaciens* [10].

Tde1 and Tde2 require specific adaptor/chaperone proteins to be loaded onto their cognate VgrG spike [16]. Thus, we next examined whether secretion of the two VgrG spike proteins requires the cognate adaptor/chaperones: Tap-1 (DUF4123-containing protein) for VgrG1 and Atu3641 (DUF2169-containing protein) for VgrG2. Secretion was assayed in Δ*tap-1* and Δ*atu3641*, mutants deleted of genes encoding the adaptor/chaperones for Tde1 and Tde2, respectively (Fig 2A) [16]. Non-secreted T6SS proteins (TssB and ClpV) as well as secreted proteins (Hcp, VgrG1/2, Tde1, and Tae) accumulated to detectable levels in the cellular fractions of all strains (Figs EV1A and 2A). VgrG1 and Tde1 were only detected in the extracellular fraction of cells that encoded Tap-1. Likewise, VgrG2 was only detected in the extracellular fraction of cells that encoded Atu3641. Hcp and Tae were detected in the extracellular fraction when at least one VgrG variant was secreted from cells. Importantly, Hcp, VgrG1/2, Tde1, and Tae were not detected in the extracellular fraction of the Δ*tap-1*Δ*atu3641* mutant. Secretion of VgrG1 or VgrG2 could be restored when the mutant was complemented with *tap-1* or *atu3641*, respectively (Fig 2B). As controls, relevant T6SS proteins were detected in the cellular fractions (Figs 2B and EV1B).

As reported [10], Tde1 accumulates to lower levels in the absence of *tap-1* (Fig 2A and B). Use of the *trc* promoter in an attempt to over-express Tde1-Tdi1 or Tde2-Tdi2 from a plasmid in Δ*tap-1*Δ*atu3641* was unable to restore secretion of cognate VgrG variants (Fig 2B). Because Tde1 always accumulated at lower levels in the absence of *tap-1* regardless of whether it is expressed endogenously or from a plasmid, we could not exclude the possibility that the deficiency of

VgrG1 secretion in Δ*tap-1* might be due to the lower amounts of cellular Tde1 in the absence of Tap-1. However, the data are consistent with the role of Tap-1 in stabilizing Tde1 as reported earlier [10] and demonstrated that the adaptor/chaperone is required for secretion of the cognate VgrG spike proteins.

In our previous study, when VgrG1 variants lacking the Tde1-binding region were expressed in a *vgrG1vgrG2* double deletion mutant (Δ*G1*Δ*G2*) and at higher levels than that of native VgrG1 in wild-type C58, they were still able to mediate secretion of Hcp and Tae effector, albeit at slightly lower levels [16]. We then hypothesized that overexpression of VgrG in the absence of a cognate effector may be sufficient to initiate the assembly of the T6SS. A polymutant strain (Δ*tde1-tdi1*Δ*G1*Δ*G2op*) lacking both Tde1/2 effectors and VgrG1/2 was generated and used for overexpression of VgrG1 and its variants. As expected, Hcp did not accumulate in the extracellular fraction of this mutant (Fig EV2A). However, when wild-type VgrG1 or truncated variants (G1_812, G1_804, and G1_785) that are abrogated in their ability to interact with Tde1 were overexpressed in this mutant, Hcp could be detected at high levels in the extracellular fraction. The shortest variant of VgrG1 (G1_781) previously shown to be incapable of restoring Hcp secretion in Δ*G1*Δ*G2* [16] was unable to restore Hcp secretion in Δ*tde1-tdi1*Δ*G1*Δ*G2op* (Fig EV2A). The data demonstrated that overexpression of VgrG overrides the absence of effector genes for initiation of T6SS assembly. However, when the chromosome-encoded *vgrG1* coding sequence was truncated, variants G1_785 and G1_781 were barely detectable in the extracellular fraction while maintaining cellular Tde1 at wild-type levels (Figs 2C and EV2B). Furthermore, the defect in the secretion of G1_785 and G1_781 is correlated with the lack of detectable extracellular Hcp, Tae, and Tde1. The data altogether demonstrated that Tde1 effector loading onto VgrG1 rather than the abundance of cellular Tde1 is critical for VgrG1 secretion and further supports the importance of Tde loading onto cognate VgrG in T6SS assembly.

The absence of Tde effectors may have affected VgrG stability and compromised the efficiency of T6SS assembly because the intensities of the signals for the truncated VgrG1 (G1_785 and G1_781) variants are slightly lower than that of full-length VgrG1 (Fig 2C). Thus, we quantified the change in levels of VgrG1/2 in wild-type and Δ*tdei* strains after adding chloramphenicol, an inhibitor of translation. The levels of VgrG1/2 decayed at similar rates in the two strains (Figs EV3A and B). Therefore, the stability and

steady-state levels of VgrG1 and VgrG2 proteins do not appear to be affected by the presence or absence of their cognate Tde effector or their cognate adaptor/chaperone (Figs 1 and 2). The data above demonstrated that loading cargo effectors onto cognate VgrG proteins is important for T6SS assembly and for ejecting the Hcp tube as well as VgrG spike proteins from *A. tumefaciens* strain C58.

## The presence of Tde effectors is required for efficient TssBC sheath assembly

Given these findings, we next tested whether the presence of Tde effectors is critical to initiate Hcp polymerization and assembly of the TssBC sheath. TssB fused to a fluorescent protein has been shown to assemble into the TssBC sheaths that were observed as fluorescent foci or streaks across the cell width [6,18]. The sheaths were reported to be dynamical contractile structures that were able to contract and disassemble after the extended sheaths were formed [6,18]. Thus, these fluorescent structures under the microscope can be used as an indicator of the sheath polymerization and T6SS assembly. Therefore, a plasmid expressing TssB with a C-terminal fusion to green fluorescence protein (TssB-GFP) was expressed in the Δ*tssB*, Δ*tssL*Δ*tssB*, and Δ*tdei*Δ*tssB* mutant strains of *A. tumefaciens* C58. TssB-GFP in Δ*tssB* partially restored T6SS-mediated secretion, as determined on the basis of the amount of secreted Hcp relative to that of the Δ*tssB* expressing an unmodified variant of TssB (Fig EV4). The protein abundance of TssB-GFP in different strains was similar, as determined on the basis of band intensities of TssB-GFP and its truncated form (Fig EV4). Similar to the previous reports [6], the sheaths were often observed as foci or streaks across the cell width. Moreover, these fluorescent structures were observed in the vast majority of the Δ*tssB* (TssB-GFP) cells (Fig 3A). We then counted the number of fluorescent foci in each strain (Fig 3B). The results showed that the Δ*tssB* (TssB-GFP) strain had slightly more than one fluorescent structure per cell, and the Δ*tssB*Δ*tdei* (TssB-GFP) strain had very few fluorescent structures (2–5 foci out of 100 cells). The fluorescent streaks were rarely seen in the later strain (Fig 3A and B). In contrast, no streaks were found in the negative control strain, Δ*tssB*Δ*tssL* (TssB-GFP; Fig 3A and B).

We also isolated and characterized T6SS sheaths, which are tubular structures formed by TssB and TssC proteins [6]. Pellets and supernatants from crude protein extracts after sheath

**Figure 3. The presence of Tde effectors is required for efficient TssBC sheath assembly.**

A  Representative images of cells of *Agrobacterium tumefaciens* C58 strains Δ*tssB*, Δ*tssL*Δ*tssB*, and Δ*tdei*Δ*tssB* cells each expressing TssB-GFP from pTrc200. Upper panel: phase contrast images; lower panel: green fluorescence images. Examples of GFP streaks were indicated by white arrowheads. Scale bar: 2 μm.
B  Quantification of TssB-GFP foci in different genetic background. The number of the fluorescent foci was counted using the "analysis particle" function in Fiji with a manually set threshold. The number of cells was counted manually. For each strain, a total of three images were obtained from one independent experiment and each image contained more than 300 cells for quantification and statistical analysis. One-way ANOVA followed by Turkey HSD test was used for statistical analysis. Data are mean ± SEM, and two groups with significant differences ($P = 0.000272$) are indicated with different letters (a and b). Similar results are reproduced in two independent experiments.
C  Western blots of the isolated sheaths, which were prepared via ultracentrifugation of cell lysates from *A. tumefaciens* C58 wild-type, Δ*tssL*, and Δ*tdei*. T: total proteins from the cell lysate. P: pellet samples containing TssBC sheaths after ultracentrifugation. S: supernatant after ultracentrifugation. Proteins were analyzed in Western blots probed with indicated antibodies. Fha serves as a cytoplasmic control. Molecular weight markers (in kDa) are indicated on the left. Similar results were obtained from at least two independent experiments. Data are from one independent experiment and reproduced in at least two independent experiments.
D  Pellet samples of wild type and Δ*tdei* were visualized with transmission electron microscopy (TEM). Sheaths are indicated by red arrowheads, and flagella, which are distinguishable on the basis of smoother, more solid, and thinner structures (~ 13 nm in width), are indicated by green arrowheads [33]. Data are from one independent experiment and reproduced in at least two independent experiments. Scale bar: 0.5 μm (upper panel), 100 nm (upper panel).

Source data are available online for this figure.

                                                                          

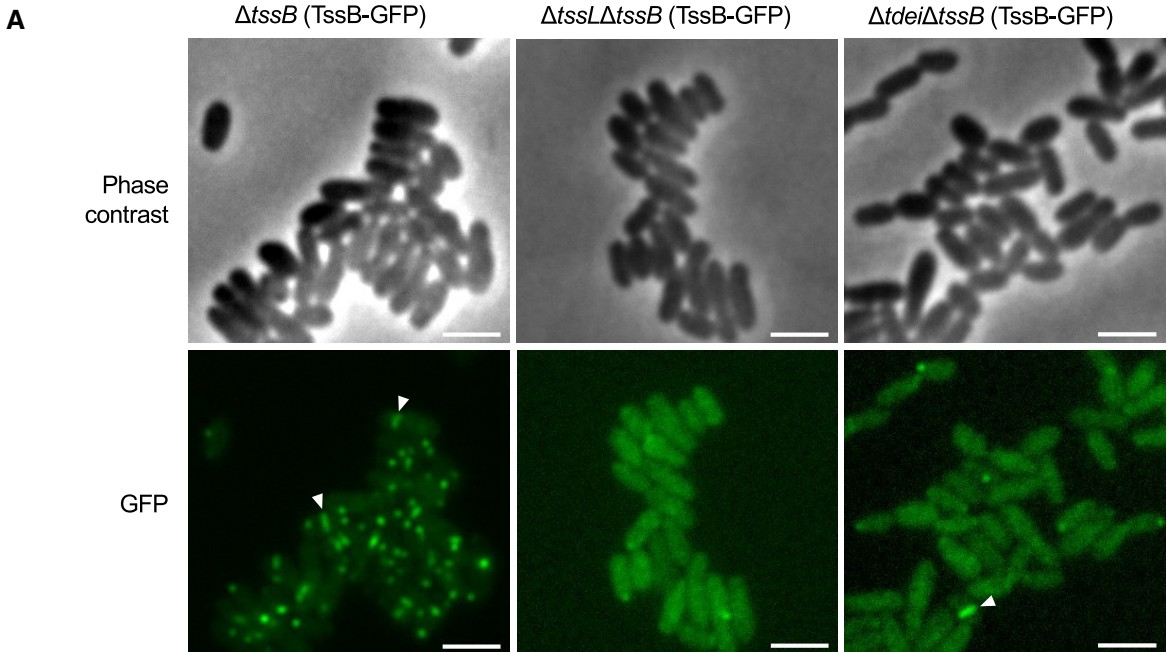

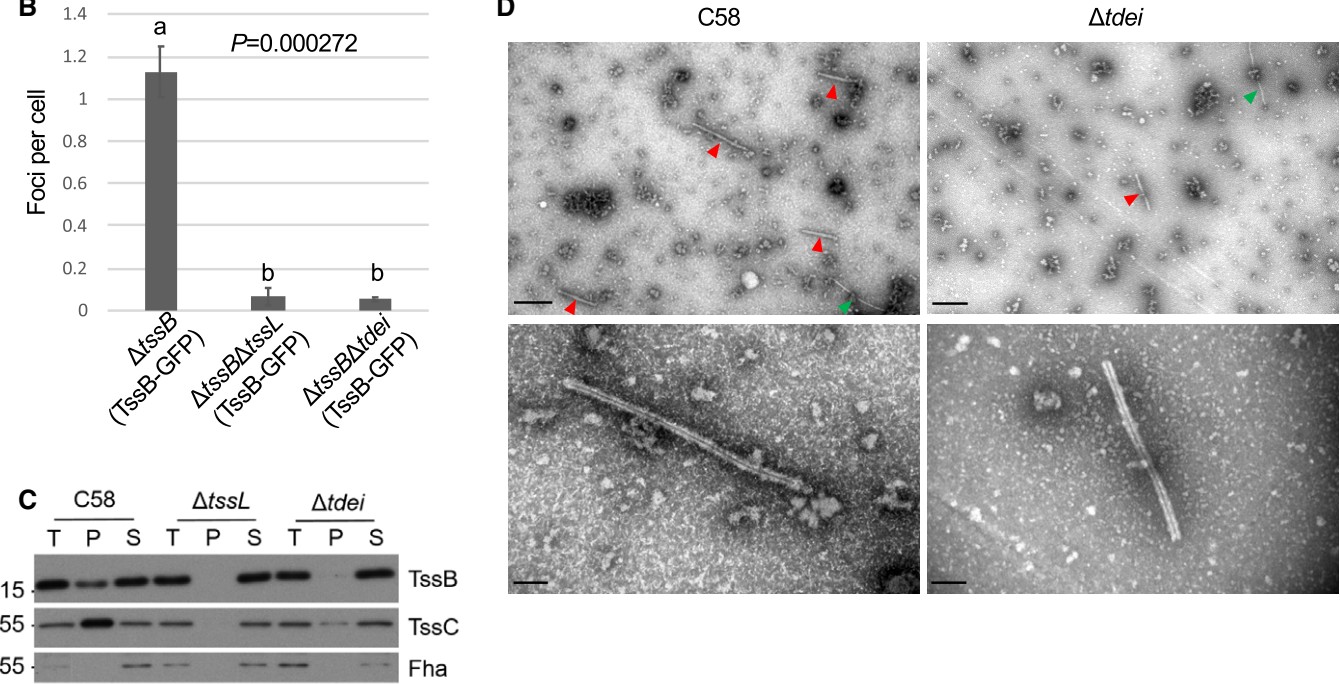

**Figure 3.**

preparation procedure of the wild-type, ∆*tssL*, and ∆*tdei* strains of *A. tumefaciens* C58 were first analyzed by Western blots (Fig 3C). The presence of TssB and TssC proteins in the pellet indicates that they have polymerized into a sheath, whereas monomers/non-polymerized subunits remain in the supernatant. TssB and TssC proteins were detected in the pellet fraction of the wild type but not of the ∆*tssL* mutant. The amounts of TssB and

TssC detected in the pellet fractions of ∆*tdei* were substantially lower than those detected in the wild type, suggesting that the TssBC sheaths were inefficiently formed in the absence of Tde effector-encoding genes.

The pellet fractions were further characterized via transmission electron microscopy (TEM). Sheath-like structures were frequently detected in fractions from C58, and only very few were observed in

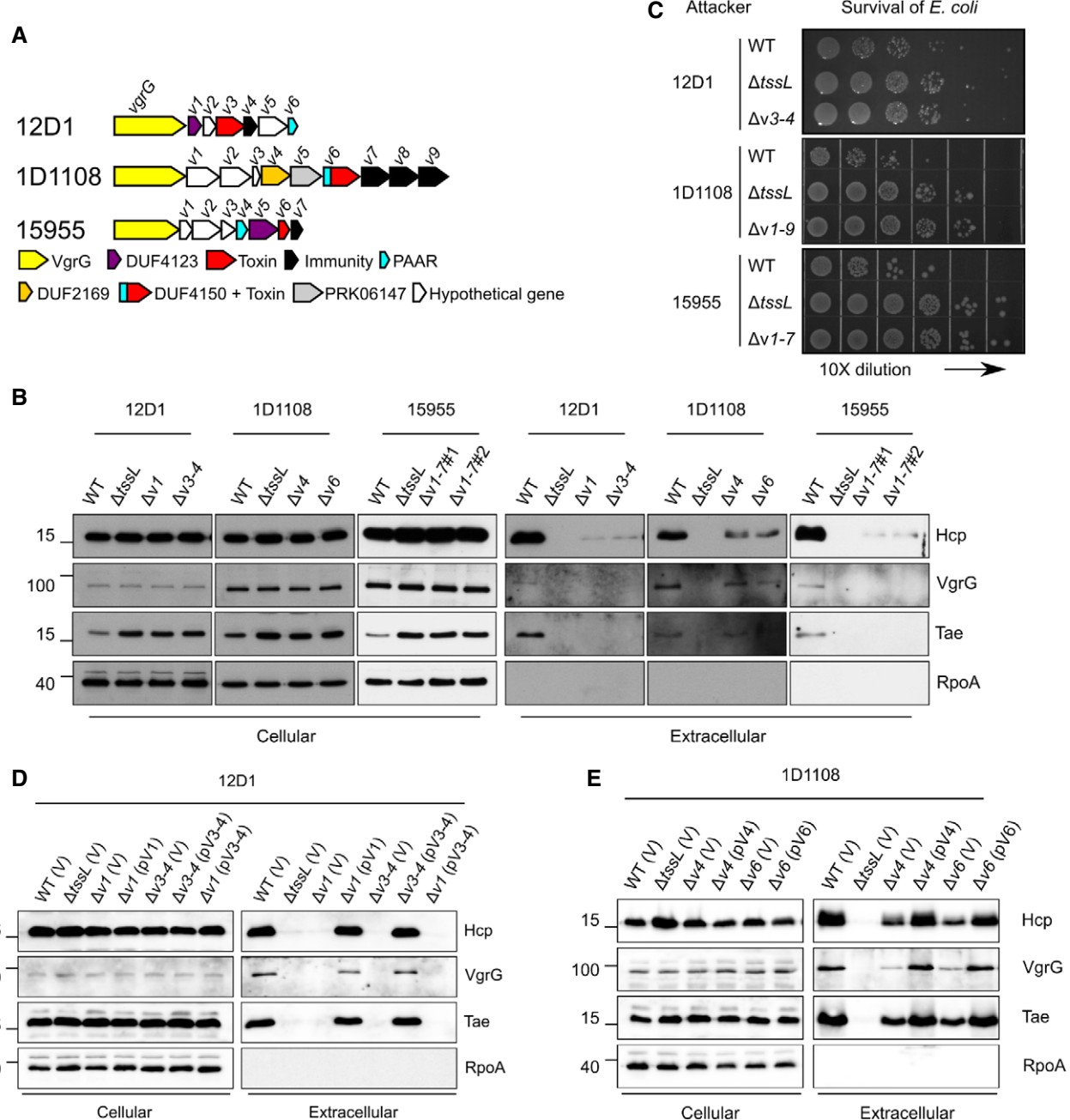

**Figure 4. Impacts of *vgrG*-associated genes on T6SS secretion and antibacterial activity of *Agrobacterium tumefaciens* strains other than C58.**

A  The *vgrG* genetic modules of the tested *A. tumefaciens* strains. Genes are color-coded according to the predicted function or results of functional assays [24]. The *vgrG*-associated genes are *v1–v6* for 12D1, *v1–v9* for 1D1108, and *v1–v7* for 15955.

B  Secretion assay of wild type and various mutants of 12D1, 1D1108, and 15955.

C  Antibacterial activity assay of wild type and mutants of 12D1, 1D1108, and 15955 was carried out in a ratio of 30:1 against *Escherichia coli* harboring the plasmid pRL662. The target *E. coli* cells were serially diluted and grown overnight on gentamicin-containing LB agar prior to photographing. Each competition was done at least four times and reproduced in three independent experiments.

D  Secretion assay of wild-type 12D1, Δ*tssL*, Δ*v1*, Δ*v3–4* containing pRL662 empty vector (V) or its derivatives with *v1* or *v3–4* were analyzed for secretion.

E  Secretion assay of wild-type 1D1108, Δ*tssL*, Δ*v4*, Δ*v6* containing pRL662 empty vector (V) or its derivatives with *v4* or *v6*.

Data information: In (B, D, E), cellular and extracellular fractions were collected from *A. tumefaciens* strains grown in liquid 523 medium. Proteins were collected and analyzed in Western blots probed with indicated antibodies. RpoA is RNA polymerase subunit alpha and located in the cytosol of *A. tumefaciens*. Molecular weight markers (in kDa) are indicated on the left. Data are from one independent experiment and reproduced in at least two independent experiments.

Source data are available online for this figure.

fractions from Δ*tdei* (Fig 3D). No sheath structures were identified from the fraction from Δ*tssL*. Regardless of the source, the sheaths were similar in structure. The diameter of the sheath structure was calculated to be ~ 30 nm. They have a hollow lumen, suggesting these were contracted sheaths which had ejected the Hcp tube. Relative to sheaths of other bacteria, those from *A. tumefaciens* have a similar morphology and diameter with the ones reported in other bacteria (25–33 nm) [6,19–22]. The data together suggest that effector-loaded VgrG is an important trigger for efficient TssBC sheath assembly. Such a mechanism may be employed by *A. tumefaciens* to prevent the assembly of the T6SS machine when effectors are not present and/or loaded.

### Effector loading onto cognate VgrG carrier is critical for Hcp/VgrG secretion in different *A. tumefaciens* genomospecies

*Agrobacterium tumefaciens* is recognized as genetically diverse and subdivided into genomospecies, which are equivalent to species-level groups [23]. We therefore tested whether loading of cargo effector onto VgrG to activate T6SS is also used by other *A. tumefaciens* genomospecies/strains. Several strains of *A. tumefaciens* encode functional T6SSs that are necessary for interbacterial competition [24]. From these, *A. tumefaciens* strains 1D1108, 15955, and 12D1, which belong to different genomospecies with only ~ 80% gene content similarity to C58, were selected to test whether effector loading onto VgrG to regulate T6SS occurs in genetically diverse *A. tumefaciens* [24]. These three strains each carry a T6SS gene cluster similar to C58 but each has only a single *vgrG* module that circumscribe several downstream genes (*v1–6* for 12D1, *v1–9* for 1D1108, *v1–7* for 15955; Fig 4A) [24]. Strains 12D1 and 15955 contain Tap-1 homolog (encoded by *v1* in 12D1 and *v5* in 15955) and 1D1108 *v4* encodes a DUF2169-containing protein, the three of which are located upstream of known or predicted toxin–immunity gene pairs (*v3–4* in 12D1, *v6–7* in 1D1108, and *v6–7* in 15955) distinct among them and C58.

Mutants lacking genes encoding predicted toxin and/or adaptor/chaperone had substantially reduced levels of Hcp and VgrG proteins in extracellular fractions, as compared to that of their respective wild-type strains (Fig 4B). Consistent with the secretion results, interbacterial competition assays showed that these mutants were as compromised as their corresponding Δ*tssL* mutants in competing with *E. coli* (Fig 4C). Expression of the adaptor or toxin/immunity genes in their respective mutants restored the ability of the cells to secrete Hcp, VgrG, and Tae to that of wild-type levels (Fig 4D and E). Expression of the *v3–4* toxin–immunity pair in the Δ*v1* adaptor/chaperone mutant of 12D1 did not restore any secretion activity. These results support the conclusion that the efficient assembly of T6SS requires an adaptor/chaperone. Thus, effector loading is a common mechanism for regulating T6SS across the *A. tumefaciens* group of bacteria and perhaps also in other bacterial species.

### Effector loading mechanisms are differentially regulated by the conserved Tae toxin

The highly diminished Hcp/VgrG secretion levels in Δ*tdei*, Δ*tde2-tdi2*Δ*tae-tai*, and Δ3TIs are surprising because in a previous study, the level of secreted Hcp in the Δ3TIs mutant was similar to that of wild-type C58 [10]. A key difference between this previous

study and the one presented here is that, in the previous study, the secretion assay was conducted in an acidic minimal medium (I-medium, pH 5.5), whereas here we used a rich medium (523) for unambiguous detection of VgrG secretion. Therefore, we carried out secretion assays for single, double, and triple toxin–immunity deletion strains in I-medium (pH 5.5). As reported previously, extracellular Hcp levels in Δ3TIs are similar to those from wild-type C58. However, Hcp was barely detected in Δ*tdei* lacking both *tde1-tdi1* and *tde2-tdi2* (Fig 5A). Expression of Tae alone or the Tae-Tai pair from a plasmid in Δ3TIs reduced Hcp

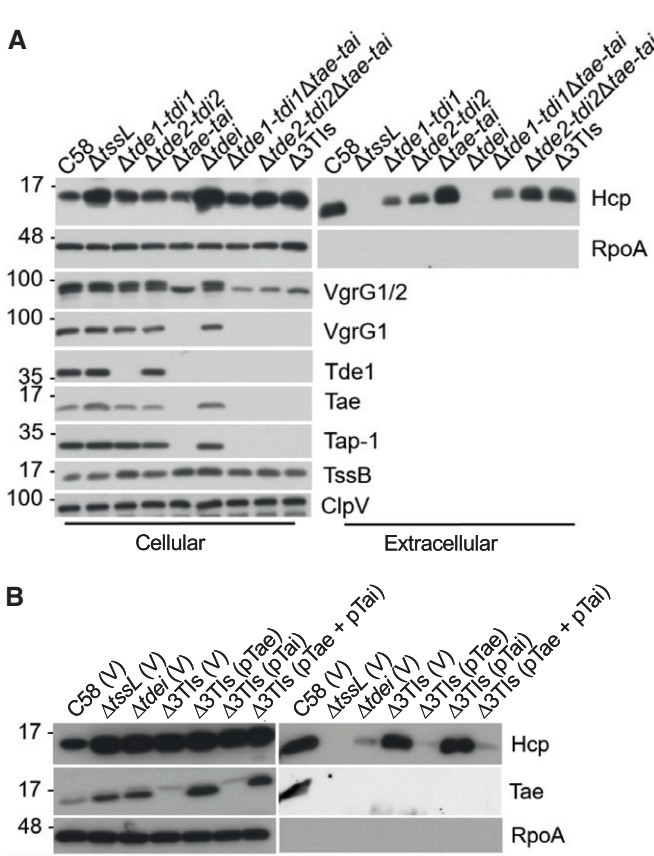

**Figure 5. Effect of *tae* on T6SS secretion and antibacterial activity in the C58 Δ*tdei* mutant grown in minimal medium.**

A   T6SS secretion assay of *Agrobacterium tumefaciens* wild-type C58, Δ*tssL*, and various single, double, and triple toxin–immunity deletion strains.

B   T6SS secretion assay of *A. tumefaciens* wild-type C58, Δ*tssL*, Δ*tdei*, and Δ3TIs harboring the indicated plasmids.

Data information: In (A, B), *A. tumefaciens* cells were grown in I-medium (pH 5.5) and cellular and extracellular fractions were collected for Western blot analysis probed with antibodies for indicated proteins. Plasmids: pRL662 vector (V), pTae (*tae* expressed pRL662), pTai (*tai* expressed on pTrc200). Molecular weight markers (in kDa) are indicated on the left. Data are from one independent experiment and reproduced in at least two independent experiments.

Source data are available online for this figure.

secretion levels, while Tai expression in Δ3TIs did not impact levels of extracellular Hcp in Δ3TIs (Fig 5B). These results suggested a role of Tae in regulating Hcp secretion levels when grown in different growth conditions.

During the course of this study, we also noticed that in Δ*tae-tai*, downstream-encoded proteins (VgrG1, Tap-1, and Tde1 proteins) were not detected in either of the cellular or extracellular fractions, while *imp* operon-encoded TssB and ClpV-encoded upstream of

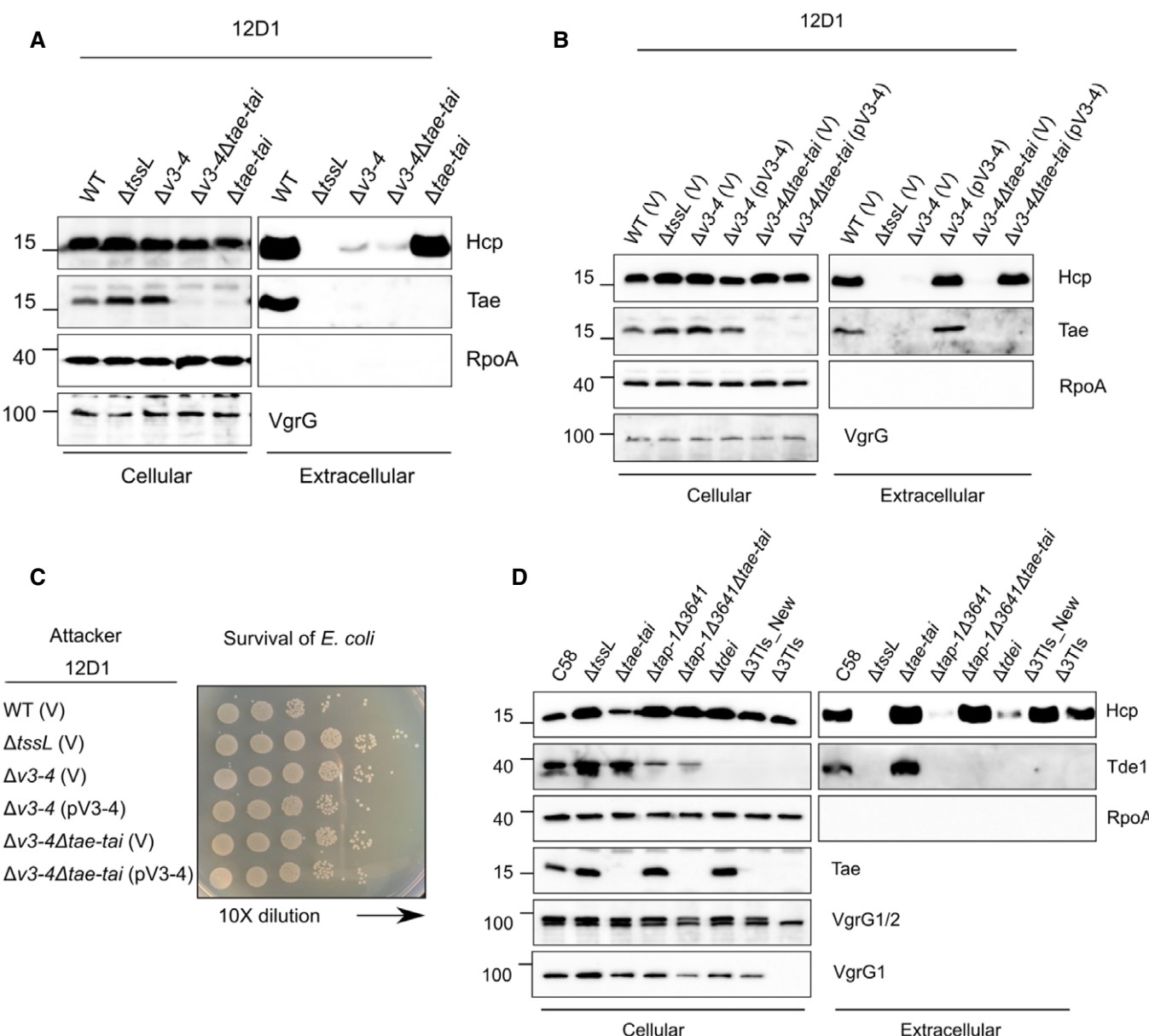

**Figure 6. Impact of *tae-tai* on T6SS secretion and antibacterial activity of *Agrobacterium tumefaciens* strains 12D1 and C58 grown in minimal medium.**

A   T6SS secretion assay of *A. tumefaciens* 12D1 wild-type, Δ*tssL*, and various toxin–immunity deletion strains.

B   T6SS secretion assay of *A. tumefaciens* 12D1 wild-type, Δ*tssL*, and various toxin–immunity deletion strains harboring the indicated plasmids.

C   Antibacterial activity assay of *A. tumefaciens* 12D1 wild-type, Δ*tssL*, and various toxin–immunity deletion strains harboring the indicated plasmids was carried out in a ratio of 30:1 against *Escherichia coli* harboring the plasmid pRL662. The target *E. coli* cells were serially diluted and grown overnight on gentamicin-containing LB agar prior to photographing. Each competition was done at least four times and reproduced in two independent experiments.

D   T6SS secretion assay of *A. tumefaciens* C58 wild-type, Δ*tssL*, and various mutant strains.

Data information: In (A, B, D), *A. tumefaciens* cells were grown in I-medium (pH 5.5) and cellular and extracellular fractions were collected for Western blot analysis probed with antibodies for indicated proteins. Plasmids: pRL662 vector (V) and derivative expressing *v3–v4*. Molecular weight markers (in kDa) are indicated on the left. Data are from one independent experiment and reproduced in at least two independent experiments.

Source data are available online for this figure.

*tae-tai* were detected in all analyzed strains (Figs 1B and EV1C). This suggests that Δ*tae-tai* has a polar effect and explains why only VgrG2 but not VgrG1 was detected in Δ*tae-tai* and Δ*tde2-tdi2*Δ*tae-tai* mutants. To investigate whether the differential role of Tae in regulating Hcp secretion in different growth media also occurs in another *A. tumefaciens* strain, we made a deletion of *tae-tai* gene pair in both WT and Δ*v3–v4* of 12D1 to determine whether the loss of *tae-tai* can impact Hcp secretion in I-medium. As expected, Hcp secretion was highly reduced in Δ*v3–v4* as compared to that of WT and Δ*tae-tai*, which supports the requirement of VgrG cargo effector for efficient Hcp secretion in both rich and minimal media (Fig 6A). To our surprise, deletion of *tae-tai* from Δ*v3–v4* (Δ*v3–v4*Δ*tae-tai*) of 12D1 did not restore Hcp secretion, whereas complementing *v3–v4* in both Δ*v3–v4* and Δ*v3–v4*Δ*tae-tai* strains with *v3–v4* restored Hcp secretion to wild-type levels (Fig 6B). Accordingly, antibacterial activity against *E. coli* was correlated with active and wild-type levels of Hcp secretion (Fig 6C). In addition, the Δ*tap-1*Δ*3641* mutant of C58 with deletions of both adaptor genes for Tde1 and Tde2 effectors also remained deficient in Hcp secretion when grown in minimal medium (Fig 6D). These results reinforced the requirement of cargo effector loading onto VgrG for assembling the T6SS and ejecting toxin effector for killing competing bacterial cells in different growth environments.

Interestingly, deletion of the *tae-tai* gene pair did not cause a polar effect in 12D1 as the VgrG protein was detected at wild-type levels in Δ*tae-tai* and Δ*v3–v4*Δ*tae-tai* mutants (Fig 6A and B). This result motivated us to revisit the polar effect of Δ*tae-tai* and Δ3TIs observed in C58; *tae* was truncated in these strains and left only 20 nucleotides upstream of the 3′ *vgrG* gene (Fig EV5). We generated a new *tae-tai* deletion in the WT, Δ*tdei,* and Δ*tap-1*Δ*3641* background of C58, designated as Δ*tae-tai**, Δ3TIs*, and Δ*tap-1*Δ*3641*Δ*tae-tai**, respectively. This mutant was truncated in *tae* and left 77 nucleotides upstream of *vgrG* (Fig EV5). The newly constructed truncation in Δ*tae-tai**, Δ3TIs*, and Δ*tap-1*Δ*3641*Δ*tae-tai** did not have polar effects as VgrG1 and Tde1 proteins were detected, which, however, retained the ability to secrete Hcp (Fig 6D).

The above evidence indicates that the presence of Tae toxin alone or Tae-Tai could have different impacts on T6SS assembly and subsequent Hcp secretion under different scenario. In C58, when *A. tumefaciens* is grown in rich medium, T6SS is only assembled when the cargo effector is loaded onto its cognate VgrG regardless of the presence or absence of Tae effector. In minimal medium, the loading of cargo effector onto its VgrG is irrelevant for T6SS assembly if Tae is absent (e.g., Δ3TIs). However, in the presence of Tae or Tae-Tai pairs, which are highly conserved across *A. tumefaciens* genomspecies [24], the Tae effector ensures that the T6SS is

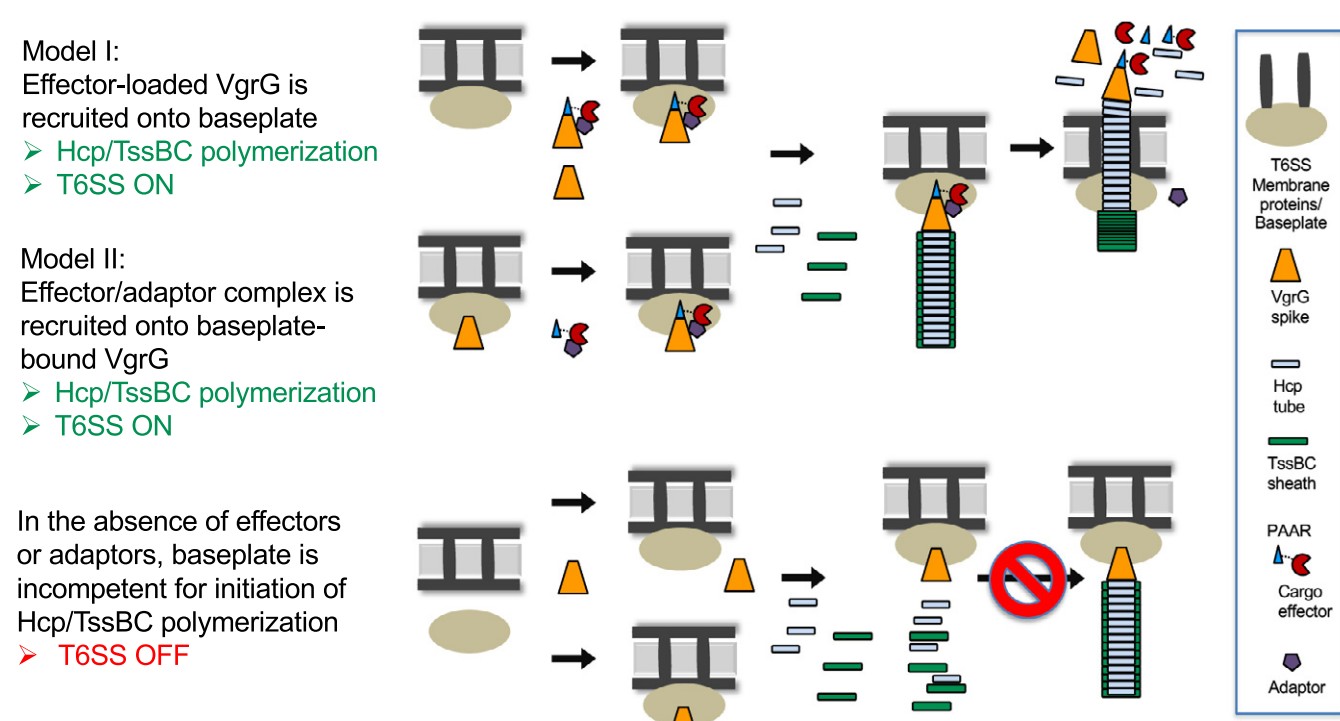

**Figure 7. Proposed models of T6SS effector loading mechanisms.**

Model I: the VgrG-PAAR–effector–adaptor complex forms first before binding to the membrane-associated baseplate complex. Model II: VgrG binds to baseplates first prior to be loaded with the effector–adaptor complex. In both, the formation of an effector-loaded VgrG-baseplate complex is critical for initiating the polymerization of the Hcp and TssBC sheath and ejecting the effector complex outward. In the absence of either cargo effectors or adaptors, VgrG may not bind to the baseplate efficiently or the VgrG-baseplate complex is incompetent for triggering T6SS assembly. The effector shown in this model represents a PAAR/DUF4150-fused effector (such as Tde2 of C58, V6 of 1D1108) but also applicable to those where the effector and PAAR are encoded in separate genes (such as Tde1 of C58, V3 of 12D1, V6 of 15955), based on the evidence of VgrG-PAAR-Tde1-Tap-1 complex formation shown previously [16]. The adaptor protein is depicted as being released from the associated effector but the step which happens remains to be determined. The proposed models do not consider the role of Tae that may differentially regulate the outcome of effector loading in T6SS assembly.

fully assembled only when VgrG is loaded with effectors. In 12D1, the T6SS is only assembled when cargo effector is loaded onto its cognate VgrG regardless of the presence or absence of Tae-Tai pair either grown in rich or minimal media. Thus, effector loading mechanisms are differentially regulated by the conserved Tae toxin. The rationale of Tae effector with different impacts in Hcp secretion under different scenario and the underlying mechanisms of how Tae effector controls T6SS assembly are beyond the scope of this study and await future investigation.

# Discussion

In this study, we describe a new mechanism that applies across the *A. tumefaciens* group of bacteria. Cargo effector loading onto its cognate VgrG is critical for T6SS assembly and ejecting toxin effectors to kill competitor bacterial cells. Such a mechanism may be an energy-saving strategy deployed by bacteria to ensure efficacious T6SS firing. However, this mechanism may have been overlooked in many other species of bacteria, because of the lack of mutants deleted of all effector-encoding genes. Likewise, in other species, the two functions of VgrG, delivering effectors and being a structural feature of the T6SS, have not been uncoupled. A recent study in *Vibrio cholerae* independently reported that recruiting effector onboard is crucial for T6SS assembly [25]. Thus, regulation of the T6SS via effector loading onto VgrG is potentially a widespread mechanism, which may be deployed by many T6SS-possessing bacteria to influence their fitness and composition of their communities. In addition, the presence of Tde2 variant with catalytic site mutation retains the ability to trigger Hcp secretion in this study is consistent with the findings that physical presence but not the enzymatic activity of three *V. cholerae* T6SS effectors is required for T6SS assembly [25]. These studies together demonstrated it is the effector loading rather than its enzymatic activity is the trigger in T6SS assembly.

Previous studies provided evidence that VgrG interacts with components of the T6SS baseplate and the interactions are critical for assembly of the T6SS [7,8,26,27]. Our findings further suggest that effector loading onto VgrG spike is the key in completing T6SS assembly. The current knowledge led us to propose hypothetical models illustrating the effector loading mechanisms (Fig 7). Model I depicts that effector-loaded VgrG exhibits higher affinity than VgrG itself for recruitment onto the T6SS baseplate complex therefore having a higher chance to trigger the T6SS assembly. Alternatively, effector loading onto the VgrG-baseplate complexes may be the trigger for Hcp polymerization and/or TssBC sheath assembly (Model II). In the absence of either cargo effectors or adaptors, VgrG may not be loaded onto the baseplate or the VgrG-baseplate complex is incapable of triggering T6SS assembly. This new concept opens future research to investigate how widespread of the T6SS effector loading mechanisms across different bacterial groups and the underlying mode of action.

# Materials and Methods

### Bacterial strains, growth conditions, and molecular techniques

Bacterial strains and sequences of primers used in this study are listed in Appendix Tables S1 and S2, respectively. *Agrobacterium tumefaciens* was grown in 523 medium at 25°C, and *E. coli* was grown in LB medium at 37°C, unless otherwise indicated [10,16,17]. Antibiotics and concentrations used were as follows: gentamycin (50 µg/ml for *A. tumefaciens* and 30 µg/ml for *E. coli*) and spectinomycin (200 µg/ml).

### DNA preparation and plasmids construction

Plasmid DNA was extracted using Presto Mini Plasmids Kit (Geneaid, Taiwan). 2× manufacturer instructions were followed in using Ready Mix A (Zymeset, Taiwan) for polymerase chain reactions (PCRs). For the construction of plasmids used for mutant generation, ~ 500 bp of the upstream and downstream sequence of gene targeted for deletion/truncation were amplified from genomic DNA with the primer sets and cloned into *Xba*I- and *Bam*HI-digested pJQ200KS plasmid. For the construction of pTrc-TssB-GFP, *A. tumefaciens tssB* and the upstream sequences corresponding to the ribosome binding site were amplified from pTrc-TssB (EML4043; Appendix Table S1) with primers *tssB_Xma*I_F and *tssB-GFP*_R (Appendix Table S2). The *gfp* gene was amplified from pBBR1-GFP (EML3) [28] using primers *GFP*_HindIII_R and *tssB-GFP*_F. The two amplified products were fused together in a second PCR that used primers *tssB_Xma*I_F and *GFP_Hind*III_R, which include restriction site sequences for *Xma*I and *Hind*III. The purified fusion product and pTrc200 plasmid were digested with *Xma*I and *Hind*III-HF (New England Biolabs, Ipswich, USA) and subsequently ligated together using T4 DNA ligase (New England Biolabs, Ipswich, USA). For plasmid pV3–4 (12D1), the PCR product of *v3–4* from 12D1 was digested with *Eco*RI and *Bam*HI, ligated to pRL662 digested with the same restriction enzymes, and transformed into *E. coli* strain DH10B. The following plasmids were generated by Gateway cloning. The PCR products of *tde2-tdi2*, *v1* from 12D1, and *v4* and *v6* from 1D1108 were recombined to pDONR222 vector via Gateway BP reaction to generate entry clones (Invitrogen, Carlsbad, CA). Constructs were verified via PCR. The genes from the entry clones were recombined via Gateway LR reaction into pRL662_RfC.1 or pTrc200_RfC.1 and transformed into *E. coli* strain DH10B. All the plasmid construct was confirmed via colony PCR, enzyme digestion, sequencing, and Western blot of *A. tumefaciens* cells harboring the plasmid.

### Mutant construction

The pJQ200KS suicide plasmid [29] and double crossover method were used to generate in-frame deletions of *A. tumefaciens* genes [30]. In brief, cells were electroporated with suicide plasmids; transformants were selected on 523 agar plates containing gentamycin without sucrose. The Gm-resistant colonies were cultured overnight in LB broth without Gm, serially diluted and spread onto 523 agar plates containing 5% sucrose without Gm to enrich for bacterial cells that had undergone a second crossover event. The deletion mutants were confirmed via colony PCR and Western blot analyses.

### Secretion assay

Secretion assays were performed as described [16]. Briefly, *A. tumefaciens* strains were cultured in 523 medium overnight and subcultured, using an initial cell density of $OD_{600 \text{ nm}} = 0.2$, in I-medium

(pH 5.5) or 523 medium, depending on the design of the experiments. After 6 h of subculturing, the samples were centrifuged at 10,000 $g$ for 10 min to separate the extracellular and the cellular fractions. The cell pellets were adjusted to $OD_{600} = 5.0$, and the respective extracellular fractions were filtered through low protein-binding 0.22 μm sterilized filter units (Millipore, Tullagreen, Ireland) and proteins were precipitated in trichloroacetic acid [15]. Western blot analyses were done as previously described [10,16,17].

**Interbacterial competition assay**

Methods for interbacterial competition assays were previously described [10]. Briefly, *A. tumefaciens* strains were co-cultured at with *E. coli* K-12 cells harboring the plasmid pRL662 (conferring gentamycin resistance), at a ratio of 30:1 on LB agar. The surviving *E. coli* cells were serially diluted, spotted, or quantified by counting colony-forming units (CFUs) on gentamycin-containing LB agar plates. Statistics were calculated using one-way ANOVA and Tukey's honestly significance difference (HSD) test (http://astatsa.com/OneWay_Anova_with_TukeyHSD/).

**Protein stability assay**

For VgrG protein stability analysis, *A. tumefaciens* cells were resuspended at $OD_{600} \sim 0.5$ in 523 medium containing chloramphenicol (100 μg/ml) to inhibit protein synthesis. Cells were harvested after 1, 2, and 3 h and adjusted to $OD_{600\ nm} = 5.0$ for Western blot analysis [31]. Intensities of specific protein bands were quantified using ImageJ software.

**T6SS sheath preparation and transmission electron microscopy (TEM)**

Isolation of T6SS sheaths was performed by the following methods previously described [6]. In brief, *A. tumefaciens* cells were cultured overnight in 5 ml 523 and subcultured into 50 ml I-medium at 25°C for 6 h. The cells were harvested and lysed for 15 min at 37°C in 4 ml of buffer containing 0.5× CelLytic B (Sigma-Aldrich, St. Louis, USA), 150 mM NaCl, 50 mM Tris pH 7.4, lysozyme (500 μg/ml), DNase I (50 μg/ml), 1 mM phenylmethylsulfonyl-fluoride (PMSF), and 0.05% Triton X-100. Cell debris was removed via centrifugation at 10,000 $g$, 10 min at 4°C. The high-molecular-weight complexes were separated from the clear cell lysate via ultracentrifugation at 150,000 $g$ for 1 h at 4°C. Pellet fractions were resuspended in 150 μl buffer containing 150 mM NaCl, 50 mM Tris pH 7.4, and 0.75 μl Protease Inhibitor Cocktail Set III (EMD Millipore, Pacific Center Court San Diego, USA). The samples were analyzed via Western analyses. For TEM, 10 μl of the samples was deposited on copper grids with carbon-formvar film support. After 3 min, samples were negative stained for 90 seconds with 25% uranyl acetate. A Tecnai G2 Spirit TEM (FEI, Hillsboro, USA) set at 80 kV was used to visualize samples.

**Fluorescence microscopy**

Cells were cultured in 523 medium overnight followed by subculturing in I-medium for 3 h. A total of 1 ml of each bacterial culture was fixed with 8 μl 25% glutaraldehyde and 125 μl 37% formaldehyde for 20 min and washed twice with PBS (Biomate, Taiwan)

with additional 0.5% Tween-20 (PBST). Bacterial cells were resuspended with 30 μl PBST. Three microliter of the fixed cell suspension was deposited onto agarose pad (PBST with 2% agarose) to reduce random movement of cells. An upright microscope BX61 (Olympus, Tokyo, Japan) equipped with a CMOS camera (C11440 ORCA-R2 Flash 2.8, Hamamatsu, Japan), objective lens (UPlanFLN 100×/1.30, Olympus, Tokyo, Japan), and a GFP filter set (Part number: FITC-3540C-000, Semwork, New York, USA) were used. Images were acquired and processed using Improvision Volocity 6.3 software (Perkin Elmer, Waltham, USA) and Fiji [32], respectively.

**Expanded View** for this article is available online.

## Acknowledgements

The authors thank Romain Kooger in the Institute of Molecular Biology & Biophysics, ETH Zürich, for discussions and providing the protocol for preparing TssBC sheaths. We acknowledge the assistance by Claudia Parada of the Institute of Biological Chemistry and staff in the Plant Cell Biology Core Laboratory as well as DNA Sequencing Core Laboratory at the Institute of Plant and Microbial Biology, Academia Sinica, Taiwan. The authors also thank Lay-Sun Ma, Chih-Horng Kuo, and Romain Kooger for critically reading the manuscript, Nia Santo and Manda Yu for assistance in preparing the revision, and the members of Lai laboratory for their discussions and suggestions. Funding for this project was provided by the Ministry of Science and Technology of Taiwan (MOST; grant no. 104-2311-B-001-025-MY3 and 107-2311-B-001-019-MY3 to E-ML and 106-2311-B-001-009 to Y-LS). Work in the Chang laboratory is supported in part by the National Institute of Food and Agriculture, US Department of Agriculture award 2014-51181-22384. MP is supported by the Swiss National Science Foundation (grant no. 179255) and the European Research Council (grant no. 679209). The funders had no role in study design, data collection and interpretation, or the decision to submit the work for publication.

## Author contributions

C-FW, Y-WL, and E-ML conceived and designed the experiments. Y-WL, C-FW, and J-SL performed the experiments and analyzed the data. DB, MP, and Y-LS provided the tools. JHC, Y-LS, MP, and E-ML supervised the execution of the experiments. Y-WL, E-ML, and JHC, with contributions from C-FW, J-SL, MP, and Y-LS, wrote the paper. All authors read and approved the paper.

## Conflict of interest

The authors declare that they have no conflict of interest.

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
