## [Review Process File · EMBO Reports]

Effector loading onto the VgrG carrier activates type VI secretion system assembly

Chih-Feng Wu, Yun-Wei Lien, Devanand Bondage, Jer-Sheng Lin, Martin Pilhofer, Yu-Ling Shih, Jeff H. Chang, and Erh-Min Lai

Review timeline:

Submission date:	20 February 2019
Editorial Decision:	28 March 2019
Revision received:	3 July 2019
Editorial Decision:	9 August 2019
Revision received:	12 November 2019
Accepted:	15 November 2019

Editor: Achim Breiling

Transaction Report:

1st Editorial Decision

28 March 2019

Thank you for the submission of your research manuscript to EMBO reports. We have now received reports from the three referees that were asked to evaluate your study, which can be found at the end of this email.

As you will see, all referees think the manuscript is of interest, but requires major revisions to allow publication of the study in EMBO reports. As the reports are below, I will not further detail them here. However, I feel that all points should be addressed experimentally in a revised manuscript.

Given the constructive referee comments, we would like to invite you to revise your manuscript with the understanding that all referee concerns must be addressed in the revised manuscript and in a detailed point-by-point response. Acceptance of your manuscript will depend on a positive outcome of a second round of review. It is EMBO reports policy to allow a single round of revision only and acceptance or rejection of the manuscript will therefore depend on the completeness of your responses included in the next, final version of the manuscript.

Revised manuscripts should be submitted within three months of a request for revision; they will otherwise be treated as new submissions. Please contact me if a 3-months time frame is not sufficient so that we can discuss the revisions further.

Supplementary/additional data: The Expanded View format, which will be displayed in the main HTML of the paper in a collapsible format, has replaced the Supplementary information. You can submit up to 5 images as Expanded View. Please follow the nomenclature Figure EV1, Figure EV2 etc. The figure legend for these should be included in the main manuscript document file in a section called Expanded View Figure Legends after the main Figure Legends section. Additional Supplementary material should be supplied as a single pdf labeled Appendix. The Appendix includes a table of content on the first page, all figures and their legends. Please follow the

nomenclature Appendix Figure Sx throughout the text and also label the figures according to this nomenclature.

For more details please refer to our guide to authors:
<http://embor.embopress.org/authorguide#manuscriptpreparation>

Please add a conflict of interest statement to the manuscript, below the author contributions.

Please also provide the abstract written in present tense.

Important: All materials and methods should be included in the main manuscript file.

See also our guide for figure preparation:
http://www.embopress.org/sites/default/files/EMBOPress_Figure_Guidelines_061115.pdf

Regarding data quantification and statistics, can you please specify, where applicable, the number "n" for how many independent experiments (biological replicates) were performed, the bars and error bars (e.g. SEM, SD) and the test used to calculate p-values in the respective figure legends. Please provide statistical testing where applicable. See:
<http://embor.embopress.org/authorguide#statisticalanalysis>

We now strongly encourage the publication of original source data with the aim of making primary data more accessible and transparent to the reader. The source data will be published in a separate source data file online along with the accepted manuscript and will be linked to the relevant figure. If you would like to use this opportunity, please submit the source data (for example scans of entire gels or blots, data points of graphs in an excel sheet, additional images, etc.) of your key experiments together with the revised manuscript. Please include size markers for scans of entire gels, label the scans with figure and panel number, and send one PDF file per figure.

- a complete author checklist, which you can download from our author guidelines (<http://embor.embopress.org/authorguide#revision>). Please insert page numbers in the checklist to indicate where the requested information can be found.
- a letter detailing your responses to the referee comments in Word format (.doc)
- a Microsoft Word file (.doc) of the revised manuscript text
- editable TIFF or EPS-formatted single figure files in high resolution (for main figures and EV figures)

I look forward to seeing a revised version of your manuscript when it is ready. Please let me know if you have questions or comments regarding the revision.

REFeree REPORTS

Referee #1:

In this work, Lien et al. demonstrate co-dependence of VgrG and their cognate cargo effectors for T6SS secretion in *Agrobacterium*. The authors show that deletion of either effectors or adapters results in loss of secretion of the adjacent VgrG. When both Tde effectors that neighbor the two VgrGs in *Agrobacterium* are deleted, no T6SS sheaths are visible and no T6SS is detected, as apparent by the lack of Hcp secretion. This is a very interesting study with a novel concept that opens questions as to the mechanism of T6SS assembly in other bacteria. Experiments are for the most part well thought of and carried out. A few questions should be addressed.

Major comments:

- 1) Lines 105-119: It is unclear why there is such a dramatic effect of the media pH on Hcp secretion. Should pH affect VgrG structure and thus its ability to initiate T6SS assembly? How can Hcp be secreted from D3TI if the main claim made by the authors is that in absence of Tdes the VgrGs

cannot initiate T6SS assembly? If VgrGs are not loaded in absence of Tdes in low pH, then how is Hcp secreted? Also, the authors should show the extracellular levels of the VgrGs in Figure S2 to determine what is the difference between the culture conditions in that respect.

2) Figure 2b: The deletions tested here are all part of operons containing PAAR. Since polar effects are plausible in these cases, all deletion strains should be complemented with the relevant effector and immunity or with the PAAR proteins to support the conclusions. Also, in strain 1D1108, the effects on Tae and VgrG secretion appear minor rather than substantial, and do not quite support the conclusion that this phenomenon is general in *Agrobacterium*.

Minor comments:

1) Lines 90-91: was the complementation done with Tde1 alone as indicated in the text, or with both Tde1-Tdi1 as indicated in Figure 1?

2) Line 152: Why do the authors refer to V1-5 in strain 12D1 if there are 6 genes downstream of VgrG?

3) Figure S3: Why wasn't secretion of VgrG truncations determined? Also, would it not be expected that the size of the truncated versions of VgrG be different (the VgrG1/2 bands appear to be of the same size for all truncations)?

4) Figure S4: anti-GFP shows a degradation product of TssB-GFP. How is it that no degradation product is detected with anti-TssB?

Referee #2:

In this manuscript Lien, Wu and co-workers address the importance of effector loading on VgrG spike proteins for T6SS assembly in *Agrobacterium*. Several reports suggest that T6SS effectors are not required for the assembly of T6SS and the results presented in this very interesting study challenge this idea.

The authors first show that the secretion of VgrG spike proteins of *Agrobacterium* (VgrG1 and 2) require the presence of their cognate Tde1 and Tde2 toxins. Interbacterial competition require the presence of a least one Tde effector. Interestingly, a mutant deleted of all Tde effectors (the "tdei mutant") is impaired in TssBC sheaths assembly, as well as in secretion of Hcp and VgrGs. All together, these data demonstrate that effectors presence is required for full T6SS function in *Agrobacterium*. The importance of effectors for efficient T6SS secretion was also determined in other *Agrobacterium* strains.

The authors further suggest that the loading of effector (and not only their presence) is required for T6SS function. This assertion needs further experimental support (see below). It is also not clear from this study to what extent is effector loading important for T6SS function (important? absolutely required?), as the overproduction of VgrG and VgrG1 variant seem to bypass the necessity for Tde effectors.

Specific comments:

1. A full demonstration that effector "loading" onto VgrG is needed for T6SS function is missing.
- One of the most convincing and clear data from this study is the phenotype of the delta tdei mutant, deleted of all Tde effectors (no Hcp secretion, no VgrG, no Tae, no sheaths assembly Fig 1a ,b ,c and Fig3).

The authors show that the secretion of VgrG1/2 requires the presence of their cognate adaptor (Fig 1e). To provide stronger evidence that effector loading on VgrG is required for T6SS function, I suggest to repeat the same secretion experiment as in Fig 1e using a double mutant delta-tap1 delta-3641 (and the complemented strain).

- Moreover, as Tap1 stabilizes Tde1 (Ma et al, 2014; lower steady state levels of Tde1 in the cellular fraction of the delta-tap1 mutant shown in Fig 1e), the defect of VgG1 secretion in the delta-tap1

may be due to the defect of Tde1 production level and not because Tde1 is not loaded on VgrG. Authors could try to overproduce Tde1 in the delta-tap1 mutant and test whether VgrG1 and Tde1 secretion are recovered or not.

- Does 3641 also stabilize Tde2? Authors may overproduce Tde1 and Tde2 in a double mutant delta-tap1/delta-3641 to test secretion as in Fig 1e.

- The authors could test as well if a double mutant deleted of VgrG1 Tde1 binding site and VgrG2 Tde2 binding site (deletions of the corresponding C-terminal regions) is still able to secrete Hcp, VgrGs and effectors. If not possible, the authors could test this for Tde1 only, for example, by constructing a deletion mutant of VgrG1 C-terminal and by testing the secretion of this VgrG1 C-terminal truncated variant. Such a VgrG mutant should not be able to perform its own secretion if effector loading on VgrG1 is necessary.

2. As shown here (Fig S3, text Line 131-138), the overexpression of VgrG1 in the absence of its cognate effector Tde1 seem to be sufficient to initiate the assembly of the T6SS. Thus the overproduction of VgrG and VgrG1 variant seem to bypass the necessity for Tde effectors. The authors should discuss this point.

Is it possible that the requirement of Tde effectors for T6SS function could be indirect? That the effectors could simply affect the stability of their cognate VgrG protein? As shown in Fig 1, it seems that the steady state levels of VgrG1 or VgrG2 proteins are not affected by the absence of Tde1 or Tde2 (or their cognate chaperone). However, authors cannot rule out that Tde effectors (Tde/adaptor complex) improve the stability of VgrG proteins. I suggest the authors to test the stability of VgrG1 and VgrG2, after translation inhibition, in the presence and in the absence of cognate effectors (C58 compared to the tdei mutant for example, using VgrG antibodies).

3. Line 151-154 and Fig 2 legend:

- Concerning the numbering of "vgrG associated genes" (V1 to Vx), it is not straightforward at first sight that VgrG is not included in the numbering (Fig2a). For clarity and to help the reader, the authors may write V1, V2, etc. on the arrow of the "vgrG-associated genes" for example, or write VgrG on the arrow so it is clear that the following gene are the V genes.

- Line 146-154. These data will be given in another paper that is "in press", but it is difficult to appreciate here the T6SS content of *A. tumefaciens* strains 1D1108, 15955 and 12D1.

Do these strains encode for similar T6SS machine than C58 except for vgrG and vgrG associated genes? Do you mean that 1D1108, 15955 and 12D1 each encode only one VgrG (with specific associated genes)? From Fig 2a it seems that VgrG1 from 12D1 and 15955 is related to C58 VgrG1? And 1D1108 VgrG2 to C58 VgrG2?

It seems so, but the authors may state this more clearly in the text. Without this information, it is difficult to appreciate how different from C658 these systems are, and thus how generalizable the requirement of effector loading for T6SS optimal function is.

4. Line 160-162; "A mutant strain harboring a plasmid with the paar gene failed to secrete Hcp to wild type levels, suggesting that the deficiency of Hcp secretion in the polymutant was due to the absence of other genes".

Isn't it possible that both PAAR and V1-7 are necessary for Hcp secretion? Please rephrase.

5. Typo: line 247: "were co-cultured at with". Please remove "at"

Referee #3:

The authors report the characterization of the process which loads effector proteins onto the tip proteins of bacterial Type 6 secretion systems (here exemplified by *Agrobacterium tumefaciens*). This process is essential for the transport of effector proteins into target bacterial cells, but as the authors state, the process is also required for the assembly of the Type 6 secretion apparatus itself. The new findings include that the lack of effectors or of loading the tip proteins with effectors (caused by site-directed mutagenesis which inhibits effector binding) seems to reduce the assembly of tubes and sheaths of the system, which the subunit proteins are still present in the cell at high amounts. The results seem to indicate that the system can only fully and functionally assemble when effectors are present and attach to the tip. Another novel, less surprising result is that secreted spike proteins, e.g. VrgG proteins, are less secreted in the absence of effectors. This indicates that firing of

the T6 machinery may only take place when the effectors are there and loaded.

The manuscript is concise and well written.

- 1.) The results shown in Fig. 1B is not clear. The Tdei mutant seems to express both VgrGs, yet in the secreted fraction, VgrG1 is not present. Since VgrG1 is not the cognate VgrG of the deleted effectors, how can this be explained and interpreted?
- 2.) Fig. 3D, can you please also show Hcp and VgrG in this Western blot? This will strengthen the claim that the tip proteins are or are not associated with the pelleted "tube" fraction.
- 3.) *A. tumefaciens* seems not to be a very efficient killer of *E. coli*. How is this explained?

1st Revision - authors' response

3 July 2019

Referee #1:

In this work, Lien et al. demonstrate co-dependence of VgrG and their cognate cargo effectors for T6SS secretion in *Agrobacterium*. The authors show that deletion of either effectors or adapters results in loss of secretion of the adjacent VgrG. When both Tde effectors that neighbor the two VgrGs in *Agrobacterium* are deleted, no T6SS sheaths are visible and no T6SS is detected, as apparent by the lack of Hcp secretion. This is a very interesting study with a novel concept that opens questions as to the mechanism of T6SS assembly in other bacteria. Experiments are for the most part well thought of and carried out. A few questions should be addressed.

Ans: Thank you very much for the positive considerations of this work and provided several valid points and constructed suggestions. We have addressed all the comments accordingly.

Major comments:

1) Lines 105-119: It is unclear why there is such a dramatic effect of the media pH on Hcp secretion. Should pH affect VgrG structure and thus its ability to initiate T6SS assembly? How can Hcp be secreted from $\Delta 3TI$ if the main claim made by the authors is that in absence of Tdes the VgrGs cannot initiate T6SS assembly? If VgrGs are not loaded in absence of Tdes in low pH, then how is Hcp secreted? Also, the authors should show the extracellular levels of the VgrGs in Figure S2 to determine what is the difference between the culture conditions in that respect.

Ans:

(1) The point raised by the reviewer also puzzled us for quite some time until we found that Hcp secretion levels are different in certain *A. tumefaciens* mutants when grown in different growth media (523 rich medium vs acidic minimal medium, I-medium, pH 5.5). When grown in 523 medium, Hcp is not secreted in the mutants lacking both Tde effectors (*Atdei*) or all effectors ($\Delta 3TIs$). However, Hcp is secreted in $\Delta 3TIs$ when grown in I-medium, in which its secretion is highly diminished in the presence of Tae effector (Fig 1B and EV2). These results suggested a role of Tae in regulating Hcp secretion levels in different growth conditions. In this work, we did not intend to study the factor/signal(s) regulating this differential secretion, which shall be an interesting topic for future investigation. Since Hcp secretion is barely detected in the absence of two Tde effectors (*Atdei*) in both I-medium and 523 medium, the role of Tde in T6SS assembly is therefore investigated.

(2) In contrast to being able to detect Hcp in the extracellular fraction of cells grown in I and 523 media, the detection of VgrG in the extracellular fraction is only unambiguous when cells are grown in 523 medium. Thus, we do not normally do western blot analyses to examine secretion of VgrG. Since the discrepancy of VgrG secretion levels between different growth conditions is indeed an interesting point, VgrG secretion was then conducted from the cells grown in 523 medium and I-medium in one set of experiments. The data showed that T6SS-dependent VgrG secretion can be only clearly demonstrated in 523 medium (Fig 2C) while only background levels of VgrG are detected in I-medium (Fig EV3B).

2) Figure 2b: The deletions tested here are all part of operons containing PAAR. Since polar effects are plausible in these cases, all deletion strains should be complemented with the relevant effector and immunity or with the PAAR proteins to support the conclusions. Also, in strain 1D1108, the effects on Tae and VgrG secretion appear minor rather than substantial, and do not quite support the conclusion that this phenomenon is general in *Agrobacterium*.

Ans:

We have performed complementation experiments in 12D1 and 1D1108 and included new data in revised manuscript. Data showed that expression of the adaptor or toxin/immunity genes in respective mutants restored secretion of Hcp, VgrG, and Tae to that of wild type level (Fig 3D, E). In contrast, expression of *v3-4* toxin-immunity pairs in 12D1 $\Delta v1$ mutant lacking putative adaptor cannot restore any secretion activity also supported the requirement of adaptor for loading effector onto cognate VgrG for efficient assembly of T6SS. The reviewer is correct that the effects of effector loading onto VgrG in Hcp and VgrG secretion levels are not regulated at the same degree (strong impact in C58 and 12D1 but less impact in 1D1108). To this end, it is not clear about the factors involved in this difference. However, different lines of data in revised manuscript clearly showed that effector loading onto VgrG is important in T6SS assembly and secretion levels. Therefore, we think this mechanism is likely pervasive in *Agrobacterium*, but nonetheless, given the great diversity of this polyphyletic group, we would not be surprised if there is variation in a lineage-dependent manner.

Minor comments:

1) Lines 90-91: was the complementation done with Tde1 alone as indicated in the text, or with both Tde1-Tdi1 as indicated in Figure 1?

Ans: Corrected as complementation by Tde1-Tdi1.

2) Line 152: Why do the authors refer to V1-5 in strain 12D1 if there are 6 genes downstream of VgrG?

Ans: Corrected as *v1-v6*.

3) Figure S3: Why wasn't secretion of VgrG truncations determined? Also, would it not be expected that the size of the truncated versions of VgrG be different (the VgrG1/2 bands appear to be of the same size for all truncations)?

Ans: The truncated VgrGs were smaller and migrated slightly faster in SDS-PAGE as shown in new figures (Fig 2C and EV3B)

4) Figure S4: anti-GFP shows a degradation product of TssB-GFP. How is it that no degradation product is detected with anti-TssB?

Ans: The detection of the smaller band (~26 kDa) by anti-GFP antibody but not anti-TssB antibody indicated that the degradation product of TssB-GFP represents GFP without the degraded N-terminal TssB.

Referee #2:

In this manuscript Lien, Wu and co-workers address the importance of effector loading on VgrG spike proteins for T6SS assembly in *Agrobacterium*. Several reports suggest that T6SS effectors are not required for the assembly of T6SS and the results presented in this very interesting study challenge this idea.

The authors first show that the secretion of VgrG spike proteins of *Agrobacterium* (VgrG1 and 2) require the presence of their cognate Tde1 and Tde2 toxins. Interbacterial competition require the presence of a least one Tde effector. Interestingly, a mutant deleted of all Tde effectors (the "tdei mutant") is impaired in TssBC sheaths assembly, as well as in secretion of Hcp and VgrGs. All together, these data demonstrate that effectors presence is required for full T6SS function in *Agrobacterium*. The importance of effectors for efficient T6SS secretion was also determined in other *Agrobacterium* strains.

The authors further suggest that the loading of effector (and not only their presence) is required for T6SS function. This assertion needs further experimental support (see below). It

is also not clear from this study to what extent is effector loading important for T6SS function (important? absolutely required?), as the overproduction of VgrG and VgrG1 variant seem to bypass the necessity for Tde effectors.

Ans: The authors appreciate the positive and constructive comments of the reviewer. We have tried our best to perform several additional experiments and addressed all the comments to substantiate the conclusions.

Specific comments:

1. A full demonstration that effector "loading" onto VgrG is needed for T6SS function is missing.

- One of the most convincing and clear data from this study is the phenotype of the delta tdei mutant, deleted of all Tde effectors (no Hcp secretion, no VgrG, no Tae, no sheaths assembly Fig 1a ,b ,c and Fig3).

The authors show that the secretion of VgrG1/2 requires the presence of their cognate adaptor (Fig 1e). To provide stronger evidence that effector loading on VgrG is required for T6SS function, I suggest to repeat the same secretion experiment as in Fig 1e using a double mutant delta-tap1 delta-3641 (and the complemented strain).

Ans: This is indeed an important point. We generated the double mutant and showed that Hcp, VgrG variants, Tde1 and Tae were no longer detectable in the extracellular fraction of the *Δtap-1Δatu3641* mutant whereas VgrG1 or VgrG2 were detectable when the mutant was complemented with either *tap-1* or *atu3641* respectively (Fig 2B).

- Moreover, as Tap1 stabilizes Tde1 (Ma et al, 2014; lower steady state levels of Tde1 in the cellular fraction of the delta-tap1 mutant shown in Fig 1e), the defect of VgG1 secretion in the delta-tap1 may be due to the defect of Tde1 production level and not because Tde1 is not loaded on VgrG. Authors could try to overproduce Tde1 in the delta-tap1 mutant and test whether VgrG1 and Tde1 secretion are recovered or not.

Ans: We used the *trc* promoter in an attempt to overexpress Tde1-Tdi1 or Tde2-Tdi2 in *Δtap-1Δatu3641* but was unable to restore secretion of cognate VgrG variants (Fig 2B). However, Tde1 always accumulated at lesser amounts in the absence of *tap-1* regardless of whether it is expressed endogenously or from a plasmid. This is consistent with the role of Tap-1 in stabilizing Tde1 as reported earlier (Ma et al., 2014) and demonstrated that the adaptor/chaperone is required for secretion of the cognate VgrG spike proteins. Furthermore, endogenous VgrG1 truncated variants (G1_785 and G1_781) are not secreted while maintaining cellular Tde1 at wild type levels (Fig 2C). Thus, the data altogether demonstrated that the defect of VgrG1 secretion in *Δtap-1* is because Tde1 is not loaded onto VgrG1, not simply due to lower amounts of Tde1.

- Does 3641 also stabilize Tde2? Authors may overproduce Tde1 and Tde2 in a double mutant delta-tap1/delta-3641 to test secretion as in Fig 1e.

-Endogenous Tde2 is not detectable and only can be detected by expression of DNase mutant form Tde2* from plasmid, which is also stabilized by *Atu3641* (Bondage et al., 2016). As indicated above, attempts to overexpress Tde2-Tdi2 by *trc* promoter on a plasmid in *Δtap-1Δatu3641* was unable to restore VgrG2 secretion (Fig 2B).

- The authors could test as well if a double mutant deleted of VgrG1 Tde1 binding site and VgrG2 Tde2 binding site (deletions of the corresponding C-terminal regions) is still able to secrete Hcp, VgrGs and effectors. If not possible, the authors could test this for Tde1 only, for example, by constructing a deletion mutant of VgrG1 C-terminal and by testing the secretion of this VgrG1 C-terminal truncated variant. Such a VgrG mutant should not be able to perform its own secretion if effector loading on VgrG1 is necessary.

Ans: We generated two chromosomally encoded truncated *vgrG1* alleles and performed secretion assay. When VgrG1 truncated variants (G1_785 and G1_781) were expressed endogenously, they were barely detectable in the extracellular fraction (Fig 2C, EV3B). Importantly, the deficiency of G1_785 and G1_781 secretion is correlated with the lack of extracellular Hcp, Tae, and Tde1, supporting the importance of Tde loading onto cognate VgrG in T6SS assembly.

2. As shown here (Fig S3, text Line 131-138), the overexpression of VgrG1 in the absence of

its cognate effector Tde1 seem to be sufficient to initiate the assembly of the T6SS. Thus the overproduction of VgrG and VgrG1 variant seem to bypass the necessity for Tde effectors. The authors should discuss this point.

Is it possible that the requirement of Tde effectors for T6SS function could be indirect? That the effectors could simply affect the stability of their cognate VgrG protein? As shown in Fig 1, it seems that the steady state levels of VgrG1 or VgrG2 proteins are not affected by the absence of Tde1 or Tde2 (or their cognate chaperone). However, authors cannot rule out that Tde effectors (Tde/adaptor complex) improve the stability of VgrG proteins. I suggest the authors to test the stability of VgrG1 and VgrG2, after translation inhibition, in the presence and in the absence of cognate effectors (C58 compared to the tdei mutant for example, using VgrG antibodies).

Ans: We have performed the stability assay as suggested and data showed no significant difference of VgrG1/2 stability in the presence or absence of Tde effectors (Fig EV4).

3. Line 151-154 and Fig 2 legend:

- Concerning the numbering of "vgrG associated genes" (V1 to Vx), it is not straightforward at first sight that VgrG is not included in the numbering (Fig2a). For clarity and to help the reader, the authors may write V1, V2, etc. on the arrow of the "vgrG-associated genes" for example, or write VgrG on the arrow so it is clear that the following gene are the V genes.

Ans: Amended

- Line 146-154. These data will be given in another paper that is "in press", but it is difficult to appreciate here the T6SS content of *A. tumefaciens* strains 1D1108, 15955 and 12D1.

Do these strains encode for similar T6SS machine than C58 except for vgrG and vgrG associated genes? Do you mean that 1D1108, 15955 and 12D1 each encode only one VgrG (with specific associated genes)? From Fig 2a it seems that VgrG1 from 12D1 and 15955 is related to C58 VgrG1? And 1D1108 VgrG2 to C58 VgrG2?

It seems so, but the authors may state this more clearly in the text. Without this information, it is difficult to appreciate how different from C658 these systems are, and thus how generalizable the requirement of effector loading for T6SS optimal function is.

Ans: The reviewer is correct that the three strains 1D1108, 15955 and 12D1 each only encodes a single T6SS gene cluster with single *vgrG* and downstream genes encoding different toxin-immunity pairs from strain C58 (Wu et al., 2019, see ref 18). We have added relevant information in revised manuscript.

4. Line 160-162; "A mutant strain harboring a plasmid with the paar gene failed to secrete Hcp to wild type levels, suggesting that the deficiency of Hcp secretion in the polymutant was due to the absence of other genes".

Isn't it possible that both PAAR and V1-7 are necessary for Hcp secretion? Please rephrase.

Ans: The reviewer is correct that we could not exclude the possibility that both PAAR and V1-7 are required for Hcp secretion. Since we have performed complementation experiments showing that expression of the adaptor or toxin/immunity genes in respective mutants restored secretion of Hcp, VgrG, and Tae to that of wild type level in 12D1 and 1D1108, the new data are included instead of previous *paar* complementation experiment now (Fig 3D, E).

5. Typo: line 247: "were co-cultured at with". Please remove "at"

Ans: Amended

Referee #3:

The authors report the characterization of the process which loads effector proteins onto the tip proteins of bacterial Type 6 secretion systems (here exemplified by *Agrobacterium tumefaciens*). This process is essential for the transport of effector proteins into target bacterial cells, but as the authors state, the process is also required for the assembly of the Type 6 secretion apparatus itself.

The new findings include that the lack of effectors or of loading the tip proteins with effectors (caused by site-directed mutagenesis which inhibits effector binding) seems to

reduce the assembly of tubes and sheaths of the system, which the subunit proteins are still present in the cell at high amounts. The results seem to indicate that the system can only fully and functionally assemble when effectors are present and attach to the tip. Another novel, less surprising result is that secreted spike proteins, e.g. VgrG proteins, are less secreted in the absence of effectors. This indicates that firing of the T6 machinery may only take place when the effectors are there and loaded.

The manuscript is concise and well written.

Ans: Thank you very much for the positive considerations of this work.

1.) The results shown in Fig. 1B is not clear. The Tdei mutant seems to express both VgrGs, yet in the secreted fraction, VgrG1 is not present. Since VgrG1 is not the cognate VgrG of the deleted effectors, how can this be explained and interpreted?

Ans: VgrG1 is the cognate VgrG for Tde1. Therefore, the lack of VgrG1 secretion in $\Delta tdei$ lacking both *tde1-tdi1* and *tde2-tdi2* is the basis for the key findings in this study.

2.) Fig. 3D, can you please also show Hcp and VgrG in this Western blot? This will strengthen the claim that the tip proteins are or are not associated with the pelleted "tube" fraction.

Ans: A previous study showed that Hcp and VgrG are not present in purified sheath fractions; only contracted but not extended sheaths were detected (Basler et al., 2012). Thus, we did not attempt to detect Hcp and VgrG proteins in our sheath fractions.

3.) *A. tumefaciens* seems not to be a very efficient killer of *E. coli*. How is this explained?

Ans: *E. coli* K12 strain lacks T6SS and has been widely used as a "standard prey" for T6SS-mediated killing assay although *E. coli* is not likely one that is encountered by *A. tumefaciens* in its communities. Regardless, it is susceptible, and we can measure a T6SS-dependent effect; hence it is a suitable lab model for rapidly assessing T6SS activity. Since the purpose of this killing assay is to determine the T6SS-mediated antibacterial activity, we consider the current assay is sufficient to achieve this goal.

2nd Editorial Decision

9 August 2019

Thank you for the submission of your revised manuscript to our editorial offices. We have now received the reports from the two referees that were asked to re-evaluate your study, you will find below. As you will see, referee #2 now supports the publication of your study in EMBO reports. However, referee #1 has still concerns that we ask you to address in a final revised version of your manuscript. Please also provide a detailed point-by-point response regarding the remaining points of referee #1.

Further, I have these editorial requests:

- Please provide a more comprehensive and shorter title (with not more than 100 characters including spaces).
- Please provide separate call-outs and more detailed description in the text for the panels in Fig. EV1 and EV4.
- Per journal policy, we do not allow 'data not shown' (see page 7 of your manuscript). All data referred to in the paper should be displayed in the main or Expanded View figures, or the Appendix. Thus, please add these data (or change the text accordingly, if these data are not important). See: <http://www.embopress.org/page/journal/14693178/authorguide#unpublisheddata>
- Please remove any writing indicating their size from the scale bars in the microscopic images. Please indicate the size only in the respective figure legend.
- Please provide all the methods in the main manuscript text. Do not put methods in the Appendix.
- Please add page numbers to the Appendix TOC.

- As they are significantly cropped, please provide the source data (scans of entire blots) for the entire Western blot panels (main and EV figures) together with the revised manuscript. The source data will be published in separate source data files online along with the accepted manuscript, and will be linked to the relevant figures. Please include size markers, label the scans with figure and panel number, and send one PDF file per figure.

- Please also provide the WB data with panels with equal contrast, and as unmodified as possible, similar to the source data. Several WB panels are presently over-contrasted.

- Please provide a conflict of interest statement below the author contributions in the main manuscript text.

In addition I would need from you:

- a short, two-sentence summary of the manuscript
- two to three bullet points highlighting the key findings of your study
- a schematic summary figure (in jpeg or tiff format with the exact width of 550 pixels and a height of not more than 400 pixels) that can be used as a visual synopsis on our website.

REFEREE REPORTS

Referee #1:

In this revised version of their manuscript, Lien et al. provide additional experiments to address several questions regarding missing controls for their previous results. While I still think that this is an interesting study that could have important implications to the understanding of T6SS activity, I remain bothered by some results that suggest to me that the mechanism put forward by the authors to explain their results may only stand valid under very specific, rich media conditions and is not a general explanation of the system.

The main issue is the fact that in the supplementary data the authors repeatedly find that Hcp is secreted in poor media even in a strain missing the two Tde effectors (Δ -3TI) and in a strain in which Tde2 is missing and thus VgrG1 is not expressed due to a polar effect (Δ -tde2-tdi2/ Δ -tae-tai; which is effectively the same as Δ -3TI; lines 108-9). Since Hcp secretion is considered a hallmark of T6SS activity and should not be secreted in the absence of VgrGs, it seems that something else might be going on in this system (perhaps, as the authors eluded, related to some effect of Tae?). I disagree with the authors that this is a topic for future studies since it has direct effect on the conclusions of this one and cannot be ignored.

Regarding the authors reply to my request to show secretion of VgrG in Figure EV2A - while I understand that there may be some technical difficulty with VgrG antibodies, without such data one can only conclude that the system is active when all 3 effectors are missing, as indicated by the only available measure - Hcp secretion. This, as mentioned above, raises doubt as to the validity of the suggested mechanism and suggests that something is missing in the proposed model.

Another result that indicates that the differences in media are relevant to the activity of the system and the proposed mechanism (and should thus not be ignored), is that in rich media some VgrG truncated versions are still secreted (Figure EV3A) while they are not secreted in poor media.

An additional remaining minor comment is that while the authors nicely show that the effect of effector or adapter complementation restores secretion in Figure 3D-E, they did not provide complementation for deletions in the E. coli survival assay shown in 3C. Since deletions of v1-9 in strain 1D1108 and of v1-7 in strain 15955 result also in deletion of the PAAR, an essential structural component of the T6SS, this should be tested to support the authors conclusions from these results. This is especially important in the 1D1108 strain, as deletion of v4 or v6 still retain some T6SS

activity (evident by Hcp and VgrG secretion shown in 3B) as opposed to deletion of *tssL*, suggesting that these deletions should not result in complete loss of antibacterial activity of the system that is shown in 3C.

Referee #2:

In this revised manuscript, the authors performed all the requested experiments and addressed my concerns. I have no further comments.

2nd Revision - authors' response

12 November 2019

Referee #1:

In this revised version of their manuscript, Lien et al. provide additional experiments to address several questions regarding missing controls for their previous results. While I still think that this is an interesting study that could have important implications to the understanding of T6SS activity, I remain bothered by some results that suggest to me that the mechanism put forward by the authors to explain their results may only stand valid under very specific, rich media conditions and is not a general explanation of the system.

Ans: We thank this reviewer for considering this work interesting and providing comments to help us improving the quality and depth of this work. To address the concern whether finding only occurs in a specific growth condition, we further performed secretion assay not only in rich medium (523) as previously demonstrated but also in minimal medium (I-medium). Our new data showed that the *vgrG*-associated toxin-immunity pair mutant of 12D1 ($\Delta v3-v4$) is deficient in Hcp secretion when grown in I-medium (Figure 6A, B), which is correlated with the loss of antibacterial activity (Figure 6C). In addition, the C58 mutant with deletion of both adaptors for Tde1 and Tde2 effectors ($\Delta tap-1 \Delta 3641$) also remains deficient in Hcp secretion when grown in minimal medium (Figure 6D). These new data reinforced the finding that requirement of VgrG cargo effector in T6SS assembly and ejecting toxin effector for killing competing bacterial cells occur in different growth environment.

The main issue is the fact that in the supplementary data the authors repeatedly find that Hcp is secreted in poor media even in a strain missing the two Tde effectors ($\Delta 3TI$) and in a strain in which Tde2 is missing and thus VgrG1 is not expressed due to a polar effect ($\Delta tde2-tdi2/\Delta tae-tai$; which is effectively the same as $\Delta 3TI$; lines 108-9). Since Hcp secretion is considered a hallmark of T6SS activity and should not be secreted in the absence of VgrGs, it seems that something else might be going on in this system (perhaps, as the authors eluded, related to some effect of Tae?). I disagree with the authors that this is a topic for future studies since it has direct effect on the conclusions of this one and cannot be ignored.

Ans: The reviewer is correct that Hcp secretion is considered a hallmark of T6SS activity and should not be secreted in the absence of VgrGs. Indeed, Hcp secretion only occurs in the presence of VgrG in all cases we analyzed. I shall emphasize that all the toxin-immunity pair mutants capable of Hcp secretion harbor at least one *vgrG* gene. Please note that C58 $\Delta 3TIs$ and $\Delta tde2-tdi2\Delta tae-tai$ strains still expresses VgrG2, either grown in 523 medium (Figure 1B) or I-medium (Figure 5).

However, as the reviewer points out this intriguing observation that the presence or absence of Tae-Tai toxin-immunity pair can affect the requirement of VgrG cargo effectors on T6SS assembly in strain C58 when grown in I-medium. Thus, we designed and generated *Ataetai* in both 12D1 WT and $\Delta v3-v4$ strains to address the effect of *tae-tai* locus in Hcp secretion and polar effect. As shown in Figure 6 and described in revised manuscript (page 10-11, line 258-272), we showed that the loss of *tae-tai* in $\Delta v3-v4$ ($\Delta v3-v4\Delta tae-tai$) of 12D1 does not restore Hcp secretion whereas complementation of *v3-v4* in both $\Delta v3-v4$ and $\Delta v3-v4\Delta tae-tai$ is able to restore Hcp secretion to wild type levels. While this result presents differential roles of *tae-tai* locus in regulating effector loading mechanisms, the data strengthened the requirement of VgrG cargo effector in T6SS assembly and ejecting toxin effector for killing

competing bacterial cells occur in different growth environment. Results from C58 and 12D1 grown in different growth media led us to concluded that the presence of Tae toxin alone or Tae-Tai could have different impacts in T6SS assembly and subsequent Hcp secretion under different scenario (see details in page 11-12, line 283-296). In short, we propose that Tae, encoding a putative peptidoglycan amidase, may function as a gate keeper to ensure efficacious T6SS firing when effectors are loaded. Active Hcp secretion of Δ 3TIs grown in Imedium may result from deregulation of such mechanism. We believe our new data not only reinforce our key finding for recruiting effector onto VgrG carrier in activating T6SS as a general mechanism but also discover new role of Tae effector in regulating T6SS. However, the rationale of Tae effector with different impacts in Hcp secretion under different scenario remain elusive. The underlying mechanisms of how Tae effector impacts T6SS assembly is beyond the scope of this work and awaits future investigation.

Regarding the authors reply to my request to show secretion of VgrG in Figure EV2A - while I understand that there may be some technical difficulty with VgrG antibodies, without such data one can only conclude that the system is active when all 3 effectors are missing, as indicated by the only available measure - Hcp secretion. This, as mentioned above, raises doubt as to the validity of the suggested mechanism and suggests that something is missing in the proposed model.

Ans: As indicated by the reviewer above, Hcp secretion is considered as a hallmark of T6SS activity. Considering many more copies of Hcp tubes are built for each VgrG trimeric spike and effector, it is indeed technically challenging to detect secretion of effectors and VgrG proteins. To this end, it is not clear the rationale underlying the difference of VgrG secretion between different growth media. This is an interesting observation but not the focus of this work.

Another result that indicates that the differences in media are relevant to the activity of the system and the proposed mechanism (and should thus not be ignored), is that in rich media some VgrG truncated versions are still secreted (Figure EV3A) while they are not secreted in poor media.

Ans: The data shown in Figure EV3A of R1 version are secretion assay in I-medium (now as EV2 in R2 version). Since no VgrG secretion could be detected in I-medium, we could not compare the secretion levels of both full-length and truncated VgrG1 variants to the results from 523 rich medium. In 523 rich media where we can detect VgrG secretion, truncated VgrG1 variants were not secreted.

An additional remaining minor comment is that while the authors nicely show that the effect of effector or adapter complementation restores secretion in Figure 3D-E, they did not provide complementation for deletions in the E. coli survival assay shown in 3C. Since deletions of v1-9 in strain 1D1108 and of v1-7 in strain 15955 result also in deletion of the PAAR, an essential structural component of the T6SS, this should be tested to support the authors conclusions from these results. This is especially important in the 1D1108 strain, as deletion of v4 or v6 still retain some T6SS activity (evident by Hcp and VgrG secretion shown in 3B) as opposed to deletion of tssL, suggesting that these deletions should not result in complete loss of antibacterial activity of the system that is shown in 3C.

Ans: It is interesting that cargo effector loading to its cognate VgrG carrier on T6SS assembly and secretion exhibits different degrees of impacts. However, all evidence in this study indicated this is a general mechanism. A recent study in *Vibrio cholerae* independently reported that recruiting effector onboard is crucial for T6SS assembly (Liang et al., PNAS, 2019). Thus, regulation of the T6SS via effector loading onto VgrG is potentially a widespread mechanism, which may be deployed by many T6SS-possessing bacteria to influences their fitness and composition of their communities.

Referee #2:

In this revised manuscript, the authors performed all the requested experiments and addressed my concerns. I have no further comments.

Ans: We thank this reviewer for recognizing the improved quality of this revised manuscript

acceptable for publication.

Accepted

15 November 2019

Thanks for the submission of the final revised version of your manuscript. I now went through your point-by-point response, and I consider the remaining concerns of referee #1 as adequately addressed. Thus, I am very pleased to accept your manuscript for publication in the next available issue of EMBO reports. Thank you for your contribution to our journal.

Corresponding Author Name: Erh-Min Lai
Manuscript Number: EMBOR-2019-47961V3